# Qsparse-local-SGD: Distributed SGD with Quantization, Sparsification, and Local Computations

**Debraj Basu** *
Adobe Inc.
dbasu@adobe.com

**Deepesh Data**
UCLA
deepeshdata@ucla.edu

**Can Karakus** *
Amazon Inc.
cakarak@amazon.com

**Suhas Diggavi**
UCLA
suhasdiggavi@ucla.edu

## Abstract

Communication bottleneck has been identified as a significant issue in distributed optimization of large-scale learning models. Recently, several approaches to mitigate this problem have been proposed, including different forms of gradient compression or computing local models and mixing them iteratively. In this paper we propose *Qsparse-local-SGD* algorithm, which combines aggressive sparsification with quantization and local computation along with error compensation, by keeping track of the difference between the true and compressed gradients. We propose both synchronous and asynchronous implementations of *Qsparse-local-SGD*. We analyze convergence for *Qsparse-local-SGD* in the *distributed* case, for smooth non-convex and convex objective functions. We demonstrate that *Qsparse-local-SGD* converges at the same rate as vanilla distributed SGD for many important classes of sparsifiers and quantizers. We use *Qsparse-local-SGD* to train ResNet-50 on ImageNet, and show that it results in significant savings over the state-of-the-art, in the number of bits transmitted to reach target accuracy.

## 1 Introduction

Stochastic Gradient Descent (SGD) [14] and its many variants have become the workhorse for modern large-scale optimization as applied to machine learning [5, 8]. We consider the setup where SGD is applied to the *distributed* setting, where $R$ different nodes compute *local* SGD on their *own* datasets $\mathcal{D}_r$. Co-ordination between them is done by aggregating these local computations to update the overall parameter $\mathbf{x}_t$ as, $\mathbf{x}_{t+1} = \mathbf{x}_t - \frac{\eta_t}{R} \sum_{r=1}^{R} g_t^r$, where $\{g_t^r\}_{r=1}^R$ are the local stochastic gradients at the $R$ machines for a local loss function $f^{(r)}(\mathbf{x})$ of the parameters, where $f^{(r)} : \mathbb{R}^d \to \mathbb{R}$.

It is well understood by now that sending full-precision gradients, causes communication to be the bottleneck for many large scale models [4, 7, 33, 39]. The communication bottleneck could be significant in emerging edge computation architectures suggested by federated learning [1, 17, 22]. To address this, many methods have been proposed recently, and these methods are broadly based on three major approaches: (i) *Quantization* of gradients, where nodes locally quantize the gradient (perhaps with randomization) to a small number of bits [3,7,33,39,40]. (ii) *Sparsification* of gradients, *e.g.,* where nodes locally select $\text{Top}_k$ values of the gradient in absolute value and transmit these at full precision [2, 4, 20, 30, 32, 40], while maintaining errors in local nodes for later compensation. (iii) *Skipping communication rounds* whereby nodes average their models after locally updating their models for several steps [9, 10, 31, 34, 37, 43, 45].

In this paper we propose *Qsparse-local-SGD* algorithm, which combines aggressive sparsification with quantization and local computation along with error compensation, by keeping track of the difference between the true and compressed gradients. We propose both synchronous and asynchronous[2] implementations of *Qsparse-local-SGD*. We analyze convergence for *Qsparse-local-SGD* in the *distributed* case, for smooth non-convex and convex objective functions. We demonstrate that, *Qsparse-local-SGD* converges at the same rate as vanilla distributed SGD for many important classes of sparsifiers and quantizers. We implement *Qsparse-local-SGD* for ResNet-50 using the ImageNet dataset, and show that we achieve target accuracies with a small penalty in final accuracy (approximately 1 %), with about a factor of 15-20 savings over the state-of-the-art [4, 30, 31], in the total number of bits transmitted. While the downlink communication is not our focus in this paper (also in [4, 20, 39], for example), it can be inexpensive when the broadcast routine is implemented in a tree-structured manner as in many MPI implementations, or if the parameter server aggregates the sparse quantized updates and broadcasts it.

**Related work.** The use of quantization for communication efficient gradient methods has decades rich history [11] and its recent use in training deep neural networks [27, 32] has re-ignited interest. Theoretically justified gradient compression using unbiased stochastic quantizers has been proposed and analyzed in [3, 33, 39]. Though methods in [36, 38] use induced sparsity in the quantized gradients, explicitly sparsifying the gradients more aggressively by retaining $\text{Top}_k$ components, *e.g.,* $k < 1\%$, has been proposed [2, 4, 20, 30, 32], combined with error compensation to ensure that all co-ordinates do get eventually updated as needed. [40] analyzed error compensation for QSGD, without $\text{Top}_k$ sparsification and a focus on quadratic functions. Another approach for mitigating the communication bottlenecks is by having infrequent communication, which has been popularly referred to in the literature as *iterative parameter mixing* and *model averaging*, see [31, 43] and references therein. Our work is most closely related to and builds on the recent theoretical results in [4, 30, 31, 43]. [30] considered the analysis for the centralized $\text{Top}_k$ (among other sparsifiers), and [4] analyzed a distributed version with the assumption of closeness of the aggregated $\text{Top}_k$ gradients to the centralized $\text{Top}_k$ case, see Assumption 1 in [4]. [31, 43] studied local-SGD, where several local iterations are done before sending the *full* gradients, and did not do any gradient compression beyond local iterations. Our work generalizes these works in several ways. We prove convergence for the *distributed* sparsification and error compensation algorithm, without the assumption of [4], by using the perturbed iterate methods [21, 30]. We analyze non-convex (smooth) objectives as well as strongly convex objectives for the distributed case with local computations. [30] gave a proof only for convex objective functions and for centralized case and therefore without local computations[3]. Our techniques compose a (stochastic or deterministic 1-bit sign) quantizer with sparsification and local computations using error compensation; in fact this technique works for any compression operator satisfying a regularity condition (see Definition 3).

**Contributions.** We study a distributed set of $R$ worker nodes each of which perform computations on locally stored data denoted by $\mathcal{D}_r$. Consider the empirical-risk minimization of the loss function $f(\mathbf{x}) = \frac{1}{R} \sum_{r=1}^{R} f^{(r)}(\mathbf{x})$, where $f^{(r)}(\mathbf{x}) = \mathbb{E}_{i \sim \mathcal{D}_r} [f_i(\mathbf{x})]$, where $\mathbb{E}_{i \sim \mathcal{D}_r} [\cdot]$ denotes expectation[4] over a random sample chosen from the local data set $\mathcal{D}_r$. For $f : \mathbb{R}^d \to \mathbb{R}$, we denote $\mathbf{x}^* := \arg\min_{\mathbf{x} \in \mathbb{R}^d} f(\mathbf{x})$ and $f^* := f(\mathbf{x}^*)$. The distributed nodes perform computations and provide updates to the master node that is responsible for aggregation and model update. We develop *Qsparse-local-SGD*, a distributed SGD composing gradient quantization and explicit sparsification (*e.g.,* $\text{Top}_k$ components), along with local iterations. We develop the algorithms and analysis for both synchronous as well as asynchronous operations, in which workers can communicate with the master at arbitrary time intervals. To the best of our knowledge, these are the first algorithms which combine quantization, aggressive sparsification, and local computations for distributed optimization.

Our main theoretical results are the convergence analysis of *Qsparse-local-SGD* for both (smooth) non-convex objectives as well as for the strongly convex case. See Theorem 1, 2 for the synchronous case, as well as Theorem 3, 4, for the asynchronous operation. Our analysis also demonstrates natural gains in convergence that distributed, mini-batch operation affords, and has convergence similar to vanilla SGD with local iterations (see Corollary 1, 2), for both the non-convex case (with convergence rate $\sim 1/\sqrt{T}$ for fixed learning rate) as well as the strongly convex case (with convergence rate $\sim 1/T$, for diminishing learning rate), demonstrating that quantizing and sparsifying the gradient, even after local iterations asymptotically yields an almost "free" communication efficiency gain (also observed numerically in Section 5 non-asymptotically). The numerical results on ImageNet dataset implemented for a ResNet-50 architecture demonstrates that one can get significant communication savings, while retaining equivalent state-of-the art performance with a small penalty in final accuracy.

Unlike previous works, *Qsparse-local-SGD* stores the compression error of the net *local update*, which is a sum of at most $H$ gradient steps and the historical error, in the local memory. From literature [4, 30], we know that methods with error compensation work only when the evolution of the error is controlled. The combination of quantization, sparsification, and local computations poses several challenges for theoretical analysis, including (i) the analysis of impact of local iterations on the evolution of the error due to quantization and sparsification, as well as the deviation of local iterates (see Lemma 3, 4, 8, 9) (ii) asynchronous updates together with distribution compression using operators which satisfy Definition 3, including our composed (*Qsparse*) operators. (see Lemma 11-14 in appendix). Another useful technical observation is that the composition of a quantizer and a sparsifier results in a compression operator (Lemma 1, 2); see Appendix A for proofs on the same.

We provide additional results in the appendices as part of the supplementary material. These include results on the asymptotic analysis for non-convex objectives in Theorem 5, 8 along with precise statements of the convergence guarantees for the asynchronous operation Theorem 6, 7 and numerics for the convex case for multi-class logistic classification on *MNIST* [19] dataset in Appendix D, for both synchronous and asynchronous operations.

We believe that our approach for combining different forms of compression and local computations can be extended to the decentralized case, where nodes are connected over an arbitrary graph, building on the ideas from [15, 35]. Our numerics also incorporate momentum acceleration, whose analysis is a topic for future research, for example incorporating ideas from [42].

**Organization.** In Section 2, we demonstrate that composing certain classes of quantization with sparsification satisfies a certain regularity condition that is needed for several convergence proofs for our algorithms. We describe the synchronous implementation of *Qsparse-local-SGD* in Section 3, and outline the main convergence results for it in Section 3.1, briefly giving the proof ideas in Section 3.2. We describe our asynchronous implementation of *Qsparse-local-SGD* and provide the theoretical convergence results in Section 4. The experimental results are given in Section 5. Many of the proof details and additional results are given in the appendices provided with the supplementary material.

## 2 Composition of Quantization and Sparsification

In this section, we consider composition of two different techniques used in the literature for mitigating the communication bottleneck in distributed optimization, namely, quantization and sparsification. In quantization, we reduce precision of the gradient vector by mapping each of its components by a deterministic [7, 15] or randomized [3, 33, 39, 44] map to a finite number of quantization levels. In sparsification, we sparsify the gradients vector before using it to update the parameter vector, by taking its $\text{Top}_k$ components or choosing $k$ components uniformly at random, denoted by $\text{Rand}_k$, [30].

**Definition 1** (Randomized Quantizer [3, 33, 39, 44]). *We say that $Q_s : \mathbb{R}^d \to \mathbb{R}^d$ is a randomized quantizer with $s$ quantization levels, if the following holds for every $\mathbf{x} \in \mathbb{R}^d$:* (i) $\mathbb{E}_Q[Q_s(\mathbf{x})] = \mathbf{x}$; (ii) $\mathbb{E}_Q[\|Q_s(\mathbf{x})\|^2] \leq (1 + \beta_{d,s})\|\mathbf{x}\|^2$, *where $\beta_{d,s} > 0$ could be a function of $d$ and $s$. Here expectation is taken over the randomness of $Q_s$.*

Examples of randomized quantizers include (i) *QSGD* [3, 39], which independently quantizes components of $\mathbf{x} \in \mathbb{R}^d$ into $s$ levels, with $\beta_{d,s} = \min(\frac{d}{s^2}, \frac{\sqrt{d}}{s})$); (ii) *Stochastic s-level Quantization* [33, 44], which independently quantizes every component of $\mathbf{x} \in \mathbb{R}^d$ into $s$ levels between $\text{argmax}_i x_i$ and $\text{argmin}_i x_i$, with $\beta_{d,s} = \frac{d}{2s^2}$; and (iii) *Stochastic Rotated Quantization* [33], which is a stochastic quantization, preprocessed by a random rotation, with $\beta_{d,s} = \frac{2\log_2(2d)}{s^2}$.

Instead of quantizing randomly into $s$ levels, we can take a deterministic approach and round off to the nearest level. In particular, we can just take the sign, which has shown promise in [7, 27, 32].

**Definition 2** (Deterministic Sign Quantizer [7, 15]). *A deterministic quantizer $Sign : \mathbb{R}^d \rightarrow \{+1, -1\}^d$ is defined as follows: for every vector $\mathbf{x} \in \mathbb{R}^d$, $i \in [d]$, the $i$'th component of $Sign(\mathbf{x})$ is defined as $\mathbb{1}\{x_i \geq 0\} - \mathbb{1}\{x_i < 0\}$.*

As mentioned above, we consider two important examples of sparsification operators: $\text{Top}_k$ and $\text{Rand}_k$, For any $\mathbf{x} \in \mathbb{R}^d$, $\text{Top}_k(\mathbf{x})$ is equal to a $d$-length vector, which has at most $k$ non-zero components whose indices correspond to the indices of the largest $k$ components (in absolute value) of $\mathbf{x}$. Similarly, $\text{Rand}_k(\mathbf{x})$ is a $d$-length (random) vector, which is obtained by selecting $k$ components of $\mathbf{x}$ uniformly at random. Both of these satisfy a so-called "compression" property as defined below, with $\gamma = k/d$ [30]. Few other examples of such operators can be found in [30].

**Definition 3** (Sparsification [30]). *A (randomized) function $Comp_k : \mathbb{R}^d \rightarrow \mathbb{R}^d$ is called a compression operator, if there exists a constant $\gamma \in (0, 1]$ (that may depend on $k$ and $d$), such that for every $\mathbf{x} \in \mathbb{R}^d$, we have $\mathbb{E}_C[\|\mathbf{x} - Comp_k(\mathbf{x})\|_2^2] \leq (1 - \gamma)\|\mathbf{x}\|_2^2$, where expectation is taken over $Comp_k$.*

We can apply different compression operators to different coordinates of a vector, and the resulting operator is also a compression operator; see Corollary 3 in Appendix A. As an application, in the case of training neural networks, we can apply different compression operators to different layers.

**Composition of Quantization and Sparsification.** Now we show that we can compose deterministic/randomized quantizers with sparsifiers and the resulting operator is a compression operator. Proofs are given in Appendix A.

**Lemma 1** (Composing sparsification with stochastic quantization). *Let $Comp_k \in \{\text{Top}_k, \text{Rand}_k\}$. Let $Q_s : \mathbb{R}^d \rightarrow \mathbb{R}^d$ be a stochastic quantizer with parameter $s$ that satisfies Definition 1. Let $Q_s Comp_k : \mathbb{R}^d \rightarrow \mathbb{R}^d$ be defined as $Q_s Comp_k(\mathbf{x}) := Q_s(Comp_k(\mathbf{x}))$ for every $\mathbf{x} \in \mathbb{R}^d$. Then $\frac{Q_s Comp_k(\mathbf{x})}{1 + \beta_{k,s}}$ is a compression operator with the compression coefficient being equal to $\gamma = \frac{k}{d(1 + \beta_{k,s})}$.*

**Lemma 2** (Composing sparsification with deterministic quantization). *Let $Comp_k \in \{\text{Top}_k, \text{Rand}_k\}$. Let $SignComp_k : \mathbb{R}^d \rightarrow \mathbb{R}^d$ be defined as follows: for every $\mathbf{x} \in \mathbb{R}^d$, the $i$'th component of $SignComp_k(\mathbf{x})$ is equal to $\mathbb{1}\{x_i \geq 0\} - \mathbb{1}\{x_i < 0\}$, if the $i$'th component is chosen in defining $Comp_k$, otherwise, it is equal to $0$. Then $\frac{\|Comp_k(\mathbf{x})\|_1 SignComp_k(\mathbf{x})}{k}$ is a compression operator[5] with the compression coefficient being equal to $\gamma = \max\left\{\frac{1}{d}, \frac{k}{d}\left(\frac{\|Comp_k(\mathbf{x})\|_1}{\sqrt{d}\|Comp_k(\mathbf{x})\|_2}\right)^2\right\}$.*

# 3 Qsparse-local-SGD

Let $\mathcal{I}_T^{(r)} \subseteq [T] := \{1, \ldots, T\}$ with $T \in \mathcal{I}_T^{(r)}$ denote a set of indices for which worker $r \in [R]$ synchronizes with the master. In a synchronous setting, $\mathcal{I}_T^{(r)}$ is same for all the workers. Let $\mathcal{I}_T := \mathcal{I}_T^{(r)}$ for any $r \in [R]$. Every worker $r \in [R]$ maintains a local parameter $\widehat{\mathbf{x}}_t^{(r)}$ which is updated in each iteration $t$, using the stochastic gradient $\nabla f_{i_t^{(r)}}\left(\widehat{\mathbf{x}}_t^{(r)}\right)$, where $i_t^{(r)}$ is a mini-batch of size $b$ sampled uniformly in $\mathcal{D}_r$. If $t \in \mathcal{I}_T$, the sparsified error-compensated update $g_t^{(r)}$ computed on the net progress made since the last synchronization is sent to the master node, and updates its local memory $m_t^{(r)}$. Upon receiving $g_t^{(r)}$'s from every worker, master aggregates them, updates the global parameter vector, and sends the new model $\mathbf{x}_{t+1}$ to all the workers; upon receiving which, they set their local parameter vector $\widehat{\mathbf{x}}_{t+1}^{(r)}$ to be equal to the global parameter vector $\mathbf{x}_{t+1}$. Our algorithm is summarized in Algorithm 1.

## 3.1 Main Results for Synchronous Operation

All results in this paper use the following two standard assumptions. (i) **Smoothness:** The local function $f^{(r)} : \mathbb{R}^d \rightarrow \mathbb{R}$ at each worker $r \in [R]$ is $L$-smooth, i.e., for every $\mathbf{x}, \mathbf{y} \in \mathbb{R}^d$, we have $f^{(r)}(\mathbf{y}) \leq f^{(r)}(\mathbf{x}) + \langle \nabla f^{(r)}(\mathbf{x}), \mathbf{y} - \mathbf{x} \rangle + \frac{L}{2}\|\mathbf{y} - \mathbf{x}\|^2$. (ii) **Bounded second moment:** For every

**Algorithm 1** Qsparse-local-SGD

---

1: Initialize $\mathbf{x}_0 = \widehat{\mathbf{x}}_0^{(r)} = m_0^{(r)}, \; \forall r \in [R]$. Suppose $\eta_t$ follows a certain learning rate schedule.
2: **for** $t = 0$ **to** $T - 1$ **do**
3:     **On Workers:**
4:     **for** $r = 1$ **to** $R$ **do**
5:         $\widehat{\mathbf{x}}_{t+\frac{1}{2}}^{(r)} \leftarrow \widehat{\mathbf{x}}_t^{(r)} - \eta_t \nabla f_{i_t^{(r)}}\left(\widehat{\mathbf{x}}_t^{(r)}\right); i_t^{(r)}$ is a mini-batch of size $b$ sampled uniformly in $\mathcal{D}_r$
6:         **if** $t + 1 \notin \mathcal{I}_T$ **then**
7:             $\mathbf{x}_{t+1} \leftarrow \mathbf{x}_t, m_{t+1}^{(r)} \leftarrow m_t^{(r)}$ and $\widehat{\mathbf{x}}_{t+1}^{(r)} \leftarrow \widehat{\mathbf{x}}_{t+\frac{1}{2}}^{(r)}$
8:         **else**
9:             $g_t^{(r)} \leftarrow Q\,Comp_k\left(m_t^{(r)} + \mathbf{x}_t - \widehat{\mathbf{x}}_{t+\frac{1}{2}}^{(r)}\right)$, send $g_t^{(r)}$ to the master.
10:            $m_{t+1}^{(r)} \leftarrow m_t^{(r)} + \mathbf{x}_t - \widehat{\mathbf{x}}_{t+\frac{1}{2}}^{(r)} - g_t^{(r)}$
11:            Receive $\mathbf{x}_{t+1}$ from the master and set $\widehat{\mathbf{x}}_{t+1}^{(r)} \leftarrow \mathbf{x}_{t+1}$
12:         **end if**
13:     **end for**
14:     **At Master:**
15:     **if** $t + 1 \notin \mathcal{I}_T$ **then**
16:         $\mathbf{x}_{t+1} \leftarrow \mathbf{x}_t$
17:     **else**
18:         Receive $g_t^{(r)}$ from $R$ workers and compute $\mathbf{x}_{t+1} = \mathbf{x}_t - \frac{1}{R}\sum_{r=1}^R g_t^{(r)}$
19:         Broadcast $\mathbf{x}_{t+1}$ to all workers.
20:     **end if**
21: **end for**
22: **Comment:** Note that $\widehat{\mathbf{x}}_{t+\frac{1}{2}}^{(r)}$ is used to denote an intermediate variable between iterations $t$ and $t + 1$.

---

$\widehat{\mathbf{x}}_t^{(r)} \in \mathbb{R}^d, r \in [R], t \in [T]$, we have $\underset{i \sim \mathcal{D}_r}{\mathbb{E}}[\|\nabla f_i(\widehat{\mathbf{x}}_t^{(r)})\|^2] \leq G^2$, for some constant $G < \infty$. This is a standard assumption in [4, 12, 16, 23, 25, 26, 29–31, 43]. Relaxation of the uniform boundedness of the gradient allowing arbitrarily different gradients of local functions in heterogenous settings as done for SGD in [24, 37] is left as future work. This also imposes a **bound on the variance**: $\underset{i \sim \mathcal{D}_r}{\mathbb{E}}[\|\nabla f_i(\widehat{\mathbf{x}}_t^{(r)}) - \nabla f^{(r)}(\widehat{\mathbf{x}}_t^{(r)})\|^2] \leq \sigma_r^2$, where $\sigma_r^2 \leq G^2$ for every $r \in [R]$. To state our results, we need the following definition from [31].

**Definition 4** (Gap [31])**.** *Let* $\mathcal{I}_T = \{t_0, t_1, \ldots, t_k\}$*, where* $t_i < t_{i+1}$ *for* $i = 0, 1, \ldots, k-1$*. The gap of* $\mathcal{I}_T$ *is defined as* $gap(\mathcal{I}_T) := \max_{i \in [k]}\{(t_i - t_{i-1})\}$*, which is equal to the maximum difference between any two consecutive synchronization indices.*

We leverage the perturbed iterate analysis as in [21, 30] to provide convergence guarantees for *Qsparse-local-SGD*. Under assumptions (i) and (ii), the following theorems hold when Algorithm 1 is run with any compression operator (including our composed operators).

**Theorem 1** (Convergence in the smooth (non-convex) case with fixed learning rate)**.** *Let* $f^{(r)}(\mathbf{x})$ *be* $L$*-smooth for every* $i \in [R]$*. Let* $QComp_k : \mathbb{R}^d \to \mathbb{R}^d$ *be a compression operator whose compression coefficient is equal to* $\gamma \in (0, 1]$*. Let* $\{\widehat{\mathbf{x}}_t^{(r)}\}_{t=0}^{T-1}$ *be generated according to Algorithm 1 with* $QComp_k$*, for step sizes* $\eta = \frac{\widehat{C}}{\sqrt{T}}$ *(where* $\widehat{C}$ *is a constant such that* $\frac{\widehat{C}}{\sqrt{T}} \leq \frac{1}{2L}$*) and* $gap(\mathcal{I}_T) \leq H$*. Then we have*

$$\mathbb{E}\|\nabla f(\mathbf{z}_T)\|^2 \leq \left(\frac{\mathbb{E}[f(\mathbf{x}_0)] - f^*}{\widehat{C}} + \widehat{C}L\left(\frac{\sum_{r=1}^R \sigma_r^2}{bR^2}\right)\right)\frac{4}{\sqrt{T}} + 8\left(4\frac{(1-\gamma^2)}{\gamma^2} + 1\right)\frac{\widehat{C}^2 L^2 G^2 H^2}{T}. \tag{1}$$

*Here* $\mathbf{z}_T$ *is a random variable which samples a previous parameter* $\widehat{\mathbf{x}}_t^{(r)}$ *with probability* $1/RT$*.*

**Corollary 1.** *Let* $\mathbb{E}[f(\mathbf{x}_0)] - f^* \leq J^2$*, where* $J < \infty$ *is a constant,*[6] $\sigma_{max} = \max_{r \in [R]} \sigma_r$*, and* $\widehat{C}^2 = \frac{bR(\mathbb{E}[f(\mathbf{x}_0)] - f^*)}{\sigma_{max}^2 L}$*, we have*

$$\mathbb{E}\|\nabla f(\mathbf{z}_T)\|^2 \leq \mathcal{O}\left(\frac{J\sigma_{max}}{\sqrt{bRT}}\right) + \mathcal{O}\left(\frac{J^2 bRG^2 H^2}{\sigma_{max}^2 \gamma^2 T}\right). \tag{2}$$

*In order to ensure that the compression does not affect the dominating terms while converging at a rate of $\mathcal{O}\left(1/\sqrt{bRT}\right)$, we would require[7] $H = \mathcal{O}\left(\gamma T^{1/4}/(bR)^{3/4}\right)$.*

Theorem 1 is proved in Appendix B and provides non-asymptotic guarantees, where we observe that compression does not affect the first order term. The corresponding asymptotic result (with decaying learning rate), with a convergence rate of $\mathcal{O}(\frac{1}{\log T})$, is provided in Theorem 5 in Appendix B.

**Theorem 2** (Convergence in the smooth and strongly convex case with a decaying learning rate). *Let $f^{(r)}(\mathbf{x})$ be $L$-smooth and $\mu$-strongly convex. Let $QComp_k : \mathbb{R}^d \to \mathbb{R}^d$ be a compression operator whose compression coefficient is equal to $\gamma \in (0, 1]$. Let $\{\widehat{\mathbf{x}}_t^{(r)}\}_{t=0}^{T-1}$ be generated according to Algorithm 1 with $QComp_k$, for step sizes $\eta_t = 8/\mu(a+t)$ with $gap(\mathcal{I}_T) \leq H$, where $a > 1$ is such that we have $a \geq \max\{4H/\gamma, 32\kappa, H\}$, $\kappa = L/\mu$. Then the following holds*

$$\mathbb{E}[f(\overline{\mathbf{x}}_T)] - f^* \leq \frac{La^3}{4S_T}\|\mathbf{x}_0 - \mathbf{x}^*\|^2 + \frac{8LT(T+2a)}{\mu^2 S_T}A + \frac{128LT}{\mu^3 S_T}B. \tag{3}$$

*Here* (i) $A = \frac{\sum_{r=1}^R \sigma_r^2}{bR^2}$, $B = 4\left(\left(\frac{3\mu}{2} + 3L\right)\frac{CG^2H^2}{\gamma^2} + 3L^2G^2H^2\right)$, *where* $C \geq \frac{4a\gamma(1-\gamma^2)}{a\gamma - 4H}$; (ii) $\overline{\mathbf{x}}_T := \frac{1}{S_T}\sum_{t=0}^{T-1}\left[w_t\left(\frac{1}{R}\sum_{r=1}^R \widehat{\mathbf{x}}_t^{(r)}\right)\right]$, *where* $w_t = (a+t)^2$; *and* (iii) $S_T = \sum_{t=o}^{T-1} w_t \geq \frac{T^3}{3}$.

**Corollary 2.** *For $a > \max\{\frac{4H}{\gamma}, 32\kappa, H\}$, $\sigma_{max} = \max_{r \in [R]} \sigma_r$, and using $\mathbb{E}\|\mathbf{x}_0 - \mathbf{x}^*\|^2 \leq \frac{4G^2}{\mu^2}$ from Lemma 2 in [25], we have*

$$\mathbb{E}[f(\overline{\mathbf{x}}_T)] - f^* \leq \mathcal{O}\left(\frac{G^2H^3}{\mu^2\gamma^3T^3}\right) + \mathcal{O}\left(\frac{\sigma_{max}^2}{\mu^2 bRT} + \frac{H\sigma_{max}^2}{\mu^2 bR\gamma T^2}\right) + \mathcal{O}\left(\frac{G^2H^2}{\mu^3\gamma^2T^2}\right). \tag{4}$$

*In order to ensure that the compression does not affect the dominating terms while converging at a rate of $\mathcal{O}(1/(bRT))$, we would require $H = \mathcal{O}\left(\gamma\sqrt{T/(bR)}\right)$.*

Theorem 2 has been proved in Appendix B. For no compression and only local computations, i.e., for $\gamma = 1$, and under the same assumptions, we recover/generalize a few recent results from literature with similar convergence rates: (i) We recover [43, Theorem 1], which is for non-convex case; (ii) We generalize [31, Theorem 2.2], which is for a strongly convex case and requires that each worker has identical datasets, to the distributed case. We emphasize that unlike [31, 43], which only consider local computation, we combine quantization and sparsification with local computation, which poses several technical challenges (e.g., see proofs of Lemma 3, 4, 7 in Appendix B).

## 3.2 Proof Outlines

Maintain virtual sequences for every worker

$$\widetilde{\mathbf{x}}_0^{(r)} := \widehat{\mathbf{x}}_0^{(r)} \qquad \text{and} \qquad \widetilde{\mathbf{x}}_{t+1}^{(r)} := \widetilde{\mathbf{x}}_t^{(r)} - \eta_t \nabla f_{i_t^{(r)}}\left(\widehat{\mathbf{x}}_t^{(r)}\right) \tag{5}$$

Define (i) $\mathbf{p}_t := \frac{1}{R}\sum_{r=1}^R \nabla f_{i_t^{(r)}}\left(\widehat{\mathbf{x}}_t^{(r)}\right)$, $\overline{\mathbf{p}}_t := \mathbb{E}_{i_t}[\mathbf{p}_t] = \frac{1}{R}\sum_{r=1}^R \nabla f^{(r)}\left(\widehat{\mathbf{x}}_t^{(r)}\right)$;

and (ii) $\widetilde{\mathbf{x}}_{t+1} := \frac{1}{R}\sum_{r=1}^R \widetilde{\mathbf{x}}_{t+1}^{(r)} = \widetilde{\mathbf{x}}_t - \eta_t \mathbf{p}_t$, $\widehat{\mathbf{x}}_t := \frac{1}{R}\sum_{r=1}^R \widehat{\mathbf{x}}_t^{(r)}$.

*Proof outline of Theorem 1.* Since $f$ is $L$-smooth, we have $f(\widetilde{\mathbf{x}}_{t+1}) - f(\widetilde{\mathbf{x}}_t) \leq -\eta_t\langle\nabla f(\widetilde{\mathbf{x}}_t), \mathbf{p}_t\rangle + \frac{\eta_t^2 L}{2}\|\mathbf{p}_t\|^2$. With some algebraic manipulations provided in Appendix B, for $\eta_t \leq 1/2L$, we arrive at

$$\frac{\eta_t}{4R}\sum_{r=1}^R \mathbb{E}\|\nabla f(\widehat{\mathbf{x}}_t^{(r)})\|^2 \leq \mathbb{E}[f(\widetilde{\mathbf{x}}_t)] - \mathbb{E}[f(\widetilde{\mathbf{x}}_{t+1})] + \eta_t^2 L\mathbb{E}\|\mathbf{p}_t - \overline{\mathbf{p}}_t\|^2 + 2\eta_t L^2\mathbb{E}\|\widetilde{\mathbf{x}}_t - \widehat{\mathbf{x}}_t\|^2$$

$$+ 2\eta_t L^2 \frac{1}{R}\sum_{r=1}^R \mathbb{E}\|\widehat{\mathbf{x}}_t - \widehat{\mathbf{x}}_t^{(r)}\|^2. \tag{6}$$

Under Assumptions 1 and 2, we have $\mathbb{E}\|\mathbf{p}_t - \overline{\mathbf{p}}_t\|^2 \leq \frac{\sum_{r=1}^R \sigma_r^2}{bR^2}$. To bound $\mathbb{E}\|\widetilde{\mathbf{x}}_t - \widehat{\mathbf{x}}_t\|^2$ in (6), we first show (in Lemma 7 in Appendix B) that $\widehat{\mathbf{x}}_t - \widetilde{\mathbf{x}}_t = \frac{1}{R}\sum_{r=1}^R m_t^{(r)}$, i.e., the difference of the true and the virtual parameter vectors is equal to the average memory, and then we bound the local memory at each worker $r \in [R]$ below.

**Lemma 3** (Bounded Memory). *For $\eta_t = \eta$, $gap(\mathcal{I}_T) \leq H$, we have for every $t \in \mathbb{Z}^+$ that*

$$\mathbb{E}\|m_t^{(r)}\|^2 \leq 4\frac{\eta^2(1-\gamma^2)}{\gamma^2}H^2G^2. \tag{7}$$

Using Lemma 3, we get $\mathbb{E}\|\widetilde{\mathbf{x}}_t - \widehat{\mathbf{x}}_t\|^2 \leq \frac{1}{R}\sum_{r=1}^R \mathbb{E}\|m_t^{(r)}\|^2 \leq 4\frac{\eta^2(1-\gamma^2)}{\gamma^2}H^2G^2$. We can bound the last term of (6) as $\frac{1}{R}\sum_{r=1}^R \mathbb{E}\|\widehat{\mathbf{x}}_t - \widehat{\mathbf{x}}_t^{(r)}\|^2 \leq \eta^2 G^2 H^2$ in Lemma 9 in Appendix B. Putting them back in (6), performing a telescopic sum from $t = 0$ to $T - 1$, and then taking an average over time, we get

$$\frac{1}{RT}\sum_{t=0}^{T-1}\sum_{r=1}^R \mathbb{E}\|\nabla f(\widehat{\mathbf{x}}_t^{(r)})\|^2 \leq \frac{4(\mathbb{E}[f(\widetilde{\mathbf{x}}_0)]-f^*)}{\eta T} + \frac{4\eta L}{bR^2}\sum_{r=1}^R \sigma_r^2 + 32\frac{\eta^2(1-\gamma^2)}{\gamma^2}L^2G^2H^2 + 8\eta^2 L^2G^2H^2.$$

By letting $\eta = \widehat{C}/\sqrt{T}$, where $\widehat{C}$ is a constant such that $\frac{\widehat{C}}{\sqrt{T}} \leq \frac{1}{2L}$, we arrive at Theorem 1. $\qquad\square$

*Proof outline of Theorem 2.* Using the definition of virtual sequences (5), we have $\|\widetilde{\mathbf{x}}_{t+1} - \mathbf{x}^*\|^2 = \|\widetilde{\mathbf{x}}_t - \mathbf{x}^* - \eta_t \overline{\mathbf{p}}_t\|^2 + \eta_t^2\|\mathbf{p}_t - \overline{\mathbf{p}}_t\|^2 - 2\eta_t \langle \widetilde{\mathbf{x}}_t - \mathbf{x}^* - \eta_t \overline{\mathbf{p}}_t, \mathbf{p}_t - \overline{\mathbf{p}}_t \rangle$. With some algebraic manipulations provided in Appendix B, for $\eta_t \leq 1/4L$ and letting $e_t = \mathbb{E}[f(\widehat{\mathbf{x}}_t)] - f^*$, we get

$$\mathbb{E}\|\widetilde{\mathbf{x}}_{t+1} - \mathbf{x}^*\|^2 \leq \left(1 - \frac{\mu\eta_t}{2}\right)\mathbb{E}\|\widetilde{\mathbf{x}}_t - \mathbf{x}^*\|^2 - \frac{\eta_t \mu}{2L}e_t + \eta_t\left(\frac{3\mu}{2} + 3L\right)\mathbb{E}\|\widehat{\mathbf{x}}_t - \widetilde{\mathbf{x}}_t\|^2$$
$$+ \frac{3\eta_t L}{R}\sum_{r=1}^R \mathbb{E}\|\widehat{\mathbf{x}}_t - \widehat{\mathbf{x}}_t^{(r)}\|^2 + \eta_t^2\frac{\sum_{r=1}^R \sigma_r^2}{bR^2}. \tag{8}$$

To bound the 3rd term on the RHS of (63), first we note that $\widehat{\mathbf{x}}_t - \widetilde{\mathbf{x}}_t = \frac{1}{R}\sum_{r=1}^R m_t^{(r)}$, and then we bound the local memory at each worker $r \in [R]$ below.

**Lemma 4** (Memory Contraction). *For $a > 4H/\gamma$, $\eta_t = \xi/a+t$, $gap(\mathcal{I}_T) \leq H$, there exists a $C \geq \frac{4a\gamma(1-\gamma^2)}{a\gamma - 4H}$ such that the following holds for every $t \in \mathbb{Z}^+$*

$$\mathbb{E}\|m_t^{(r)}\|^2 \leq 4\frac{\eta_t^2}{\gamma^2}CH^2G^2. \tag{9}$$

A proof of Lemma 4 is provided in Appendix B and is technically more involved than the proof of Lemma 3. This complication arises because of the decaying learning rate, combined with compression and local computation. We can bound the penultimate term on the RHS of (63) as $\frac{1}{R}\sum_{r=1}^R \mathbb{E}\|\widehat{\mathbf{x}}_t - \widehat{\mathbf{x}}_t^{(r)}\|^2 \leq 4\eta_t^2 G^2 H^2$. This can be shown along the lines of the proof of [31, Lemma 3.3] and we show it in Lemma 8 in Appendix B. Substituting all these in (63) gives

$$\mathbb{E}\|\widetilde{\mathbf{x}}_{t+1} - \mathbf{x}^*\|^2 \leq \left(1 - \frac{\mu\eta_t}{2}\right)\mathbb{E}\|\widetilde{\mathbf{x}}_t - \mathbf{x}^*\|^2 - \frac{\mu\eta_t}{2L}e_t + \eta_t\left(\frac{3\mu}{2} + 3L\right)C\frac{4\eta_t^2}{\gamma^2}G^2H^2$$
$$+ (3\eta_t L)4\eta_t^2 LG^2H^2 + \eta_t^2\frac{\sum_{r=1}^R \sigma_r^2}{bR^2}. \tag{10}$$

Since (10) is a contracting recurrence relation, with some calculation done in Appendix B, we complete the proof of Theorem 2. $\qquad\square$

## 4 Asynchronous Qsparse-local-SGD

We propose and analyze a particular form of asynchronous operation where the workers synchronize with the master at arbitrary times decided locally or by master picking a subset of nodes as in federated learning [17, 22]. However, the local iterates evolve at the same rate, i.e. each worker takes the same number of steps per unit time according to a global clock. The asynchrony is therefore that updates occur after different number of local iterations but the local iterations are synchronous with respect to the global clock.[8]

In this asynchronous setting, $\mathcal{I}_T^{(r)}$'s may be different for different workers. However, we assume that $gap(\mathcal{I}_T^{(r)}) \leq H$ holds for every $r \in [R]$, which means that there is a uniform bound on the maximum delay in each worker's update times. The algorithmic difference from Algorithm 1 is that, in this case, *a subset of* workers (including a single worker) can send their updates to the master at their synchronization time steps; master aggregates them, updates the global parameter vector, and sends that only to those workers. Our algorithm is summarized in Algorithm 2 in Appendix C. We give the simplified expressions of our main results below; more precise results are in Appendix C.

**Theorem 3** (Convergence in the smooth non-convex case with fixed learning rate). *Under the same conditions as in Theorem 1 with $gap(\mathcal{I}_T^{(r)}) \leq H$, if $\{\widehat{\mathbf{x}}_t^{(r)}\}_{t=0}^{T-1}$ is generated according to Algorithm 2, the following holds, where $\mathbb{E}[f(\mathbf{x}_0)] - f^* \leq J^2$, $\sigma_{max} = \max_{r \in [R]} \sigma_r$, and $\widehat{C}^2 = bR(\mathbb{E}[f(\mathbf{x}_0)]-f^*)/\sigma_{max}^2$.*

$$\mathbb{E}\|\nabla f(\mathbf{z}_T)\|^2 \leq \mathcal{O}\left(\frac{J\sigma_{max}}{\sqrt{bRT}}\right) + \mathcal{O}\left(\frac{J^2 bRG^2}{\sigma_{max}^2 \gamma^2 T}(H^2 + H^4)\right). \tag{11}$$

*where $\mathbf{z}_T$ is a random variable which samples a previous parameter $\widehat{\mathbf{x}}_t^{(r)}$ with probability $1/RT$. In order to ensure that the compression does not affect the dominating terms while converging at a rate of $\mathcal{O}\left(1/\sqrt{bRT}\right)$, we would require $H = \mathcal{O}\left(\sqrt{\gamma}T^{1/8}/(bR)^{3/8}\right)$.*

We give a precise result in Theorem 6 in Appendix C. Note that Theorem 3 provides non-asymptotic guarantees, where compression is almost for "free". The corresponding asymptotic result with decaying learning rate, with a convergence rate of $\mathcal{O}(\frac{1}{\log T})$, is provided in Theorem 8 in Appendix C.

**Theorem 4** (Convergence in the smooth and strongly convex case with decaying learning rate). *Under the same conditions as in Theorem 2 with $gap(\mathcal{I}_T^{(r)}) \leq H$, $a > \max\{4H/\gamma, 32\kappa, H\}$, $\sigma_{max} = \max_{r \in [R]} \sigma_r$, if $\{\widehat{\mathbf{x}}_t^{(r)}\}_{t=0}^{T-1}$ is generated according to Algorithm 2, the following holds:*

$$\mathbb{E}[f(\overline{\mathbf{x}}_T)] - f^* \leq \mathcal{O}\left(\frac{G^2 H^3}{\mu^2 \gamma^3 T^3}\right) + \mathcal{O}\left(\frac{\sigma_{max}^2}{\mu^2 bRT} + \frac{H\sigma_{max}^2}{\mu^2 bR\gamma T^2}\right) + \mathcal{O}\left(\frac{G^2}{\mu^3 \gamma^2 T^2}(H^2 + H^4)\right). \tag{12}$$

*where $\overline{\mathbf{x}}_T$, $S_T$ are as defined in Theorem 2. To ensure that the compression does not affect the dominating terms while converging at a rate of $\mathcal{O}\left(1/(bRT)\right)$, we would require $H = \mathcal{O}\left(\sqrt{\gamma}(T/(bR))^{1/4}\right)$.*

We give a more precise result in Theorem 7 in Appendix C. If $\mathcal{I}_T^{(r)}$'s are the same for all the workers, then one would ideally require that the bounds on $H$ in the asynchronous setting reduce to the bounds on $H$ in the synchronous setting. This is not happening, as our bounds in the asynchronous setting are for the worst case scenario – they hold as long as $gap(\mathcal{I}_T^{(r)}) \leq H$, for every $r \in [R]$.

### 4.1 Proof Outlines

Our proofs of these results follow the same outlines of the corresponding proofs in the synchronous setting, but some technical details change significantly. This is because, in our asynchronous setting, workers are allowed to update the global parameter vector in between two consecutive synchronization time steps of other workers. For example, unlike the synchronous setting, $\widehat{\mathbf{x}}_t - \widetilde{\mathbf{x}}_t = \frac{1}{R}\sum_{r=1}^R m_t^{(r)}$ does not hold here; however, we can show that $\widehat{\mathbf{x}}_t - \widetilde{\mathbf{x}}_t$ is equal to the sum of $\frac{1}{R}\sum_{r=1}^R m_t^{(r)}$ and an additional term, which leads to potentially a weaker bound $\mathbb{E}\|\widehat{\mathbf{x}}_t - \widetilde{\mathbf{x}}_t\|^2 \leq \mathcal{O}\left(\eta_t^2/\gamma^2 G^2(H^2 + H^4)\right)$ (vs. $\mathcal{O}\left(\eta_t^2/\gamma^2 G^2 H^2\right)$ for the synchronous setting), proved in Lemma 13-14 in Appendix C. Similarly, the proof of the average true sequence being close to the virtual sequence requires carefully chosen reference points on the global parameter sequence lying within bounded steps of the local parameters. We show a bound on $\frac{1}{R}\sum_{r=1}^R \mathbb{E}\|\widehat{\mathbf{x}}_t - \widehat{\mathbf{x}}_t^{(r)}\|^2 \leq \mathcal{O}(\eta_t^2 G^2(H^2 + H^4/\gamma^2))$, which is weaker than the corresponding bound $\mathcal{O}(\eta_t^2 G^2 H^2)$ for the synchronous setting, in Lemma 11-12 in Appendix C.

## 5 Experiments

**Experiment setup:** We train ResNet-50 [13] (which has $d = 25,610,216$ parameters) on ImageNet dataset, using 8 NVIDIA Tesla V100 GPUs. We use a learning rate schedule consisting of 5 epochs of linear warmup, followed by a piecewise decay of 0.1 at epochs 30, 60 and 80, with a batch size of 256 per GPU. For experiments, we focus on SGD with momentum of 0.9, applied on the local iterations of the workers. We build our compression scheme into the Horovod framework [28].[9] We use $SignTop_k$ (as in Lemma 2) as our composed operator. In $Top_k$, we only update $k_t = \min(d_t, 1000)$ elements per step for each tensor $t$, where $d_t$ is the number of elements in the tensor. For ResNet-50 architecture, this amounts to updating a total of $k = 99,400$ elements per step. We also perform analogous experiments on the *MNIST* [19] handwritten digits dataset for softmax regression with a standard $\ell_2$ regularizer, using the synchronous operation of *Qsparse-local-SGD* with 15 workers, and

a decaying learning rate as proposed in Theorem 2, the details of which are provided in Appendix D.[10]

**Results:** Figure 1 compares the performance of $SignTop_k$-$SGD$ (which employs the 1 bit sign quantizer and the $Top_k$ sparsifier) with error compensation (SignTopK) against (i) $Top_k$ SGD with error compensation (TopK-SGD), (ii) SignSGD with error compensation (EF-SIGNSGD), and (iii) vanilla SGD (SGD). All of these are specializations of *Qsparse-local-SGD*. Furthermore, SignTopK_hL uses a synchronization period of h; same applies for other schemes. From Figure 1a, we observe that quantization and sparsification, both individually and combined, with error compensation, has almost no penalty in terms of convergence rate, with respect to vanilla SGD. We observe that SignTopK demonstrates superior performance over EF-SIGNSGD, TopK-SGD, as well as vanilla SGD, both in terms of the required number of communicated bits for achieving a certain target loss as well as test accuracy. This is because in SignTopK, we send only 1 bit for the sign of each $Top_k$ coordinate, along with its location. Observe that the incorporation of local iterations in Figure 1a has very little impact on the convergence rates, as compared to vanilla SGD with the same number of local iterations. Furthermore, this provides an added advantage over SignTopK, in terms of savings (by a factor of 6 to 8 times on average) in communication bits for achieving a certain target loss; see Figure 1b.

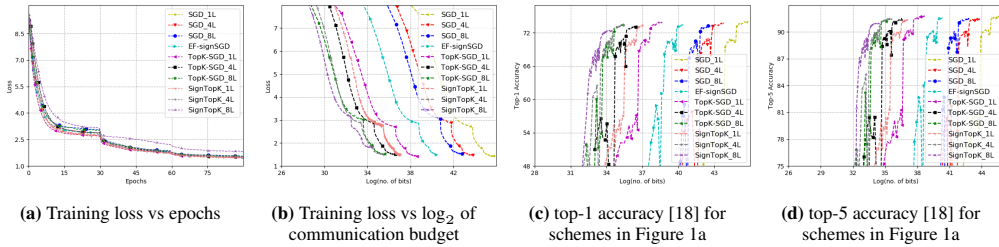

**(a)** Training loss vs epochs    **(b)** Training loss vs $\log_2$ of communication budget    **(c)** top-1 accuracy [18] for schemes in Figure 1a    **(d)** top-5 accuracy [18] for schemes in Figure 1a

**Figure 1** Figure 1a-1d demonstrate performance gains of our of our scheme in comparison with local SGD [31], EF-SIGNSGD [15] and TopK-SGD [4, 30] in a non-convex setting for synchronous updates.

Figure 1c and Figure 1d show the top-1, and top-5 convergence rates,[11] respectively, with respect to the total number of bits of communication used. We observe that *Qsparse-local-SGD* combines the bit savings of the deterministic sign based operator and aggressive sparsifier, with infrequent communication; thereby, outperforming the cases where these techniques are individually used. In particular, the required number of bits to achieve the same loss or accuracy in the case of *Qsparse-local-SGD* is around 1/16 in comparison with TopK-SGD and over 1000× less than vanilla SGD.

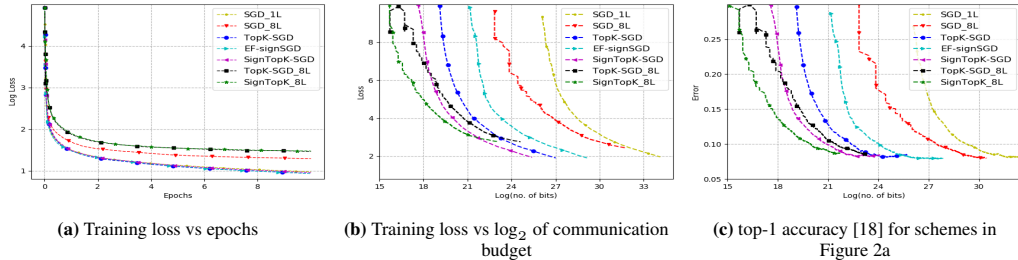

**(a)** Training loss vs epochs    **(b)** Training loss vs $\log_2$ of communication budget    **(c)** top-1 accuracy [18] for schemes in Figure 2a

**Figure 2** Figure 2a-2c demonstrate the performance gains of our scheme in a convex setting.

Figure 2b and 2c makes similar comparisons in the convex setting, and shows that for a test error approximately 0.1, *Qsparse-local-SGD* combines the benefits of the composed operator $SignTop_k$, with local computations, and needs 10-15 times less bits than TopK-SGD and 1000× less bits than vanilla SGD. Also in Figure 2a, we observe that both TopK-SGD and SignTopK_8L (SignTopK with 8 local iterations) converge at rates which are almost similar to that of their corresponding local SGD counterpart. Our experiments in both non-convex and convex settings verify that error compensation through memory can be used to mitigate not only the missing components from updates in previous synchronization rounds, but also explicit quantization error.

**Acknowledgments**

The authors gratefully thank Navjot Singh for his help with experiments in the early stages of this work. This work was partially supported by NSF grant #1514531, by UC-NL grant LFR-18-548554 and by Army Research Laboratory under Cooperative Agreement W911NF-17-2-0196. The views and conclusions contained in this document are those of the authors and should not be interpreted as representing the official policies, either expressed or implied, of the Army Research Laboratory or the U.S. Government. The U.S. Government is authorized to reproduce and distribute reprints for Government purposes notwithstanding any copyright notation here on.

## Footnotes

* Work done while Debraj Basu and Can Karakus were at UCLA.

[2]In our asynchronous model, the distributed nodes' iterates evolve at the same rate, but update the gradients at arbitrary times; see Section 4 for more details.

[3]At the completion of our work, we recently found that in parallel to our work [15] examined use of sign-SGD quantization, *without sparsification* for the centralized model. Another recent work in [16] studies the decentralized case with sparsification for strongly convex function. Our work, developed independent of these works, uses quantization, sparsification and local computations for the distributed case with local computations for both non-convex and strongly convex objectives.

[4]Our setup can also handle different local functional forms, beyond dependence on the local data set $\mathcal{D}_r$, which is not explicitly written for notational simplicity.

[5]The analysis for general $p$-norm, i.e. $\frac{\|Comp_k(\mathbf{x})\|_p SignComp_k(\mathbf{x})}{k}$, for any $p \in \mathbb{Z}_+$ is provided in Appendix A.

[6]Even classical SGD requires knowing an upper bound on $\|\mathbf{x}_0 - \mathbf{x}^*\|$ in order to choose the learning rate. Smoothness of $f$ translates this to the difference of the function values.

[7]Here we characterize the reduction in communication that can be afforded, however for a constant $H$ we get the same rate of convergence after $T = \Omega\left((bR)^3/\gamma^4\right)$. Analogous statements hold for Theorem 2-4.

[8]This is different from asynchronous algorithms studied for stragglers [26, 41], where only one gradient step is taken but occurs at different times due to delays.

[9]Our implementation is available at https://github.com/karakusc/horovod/tree/qsparselocal.

[10]Further numerics demonstrating the performance of *Qsparse-local-SGD* for the composition of a stochastic quantizer with a sparsifier, as compared to $SignTop_k$ and other standard baselines can be found in [6].

[11]top-i refers to the accuracy of the top i predictions by the model from the list of possible classes; see [18].

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
