[Supplementary Material]

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

[12] The $m$-norm of a vector $\mathbf{u} \in \mathbb{R}^d$ is defined as $\|\mathbf{u}\|_m := \left(\sum_{i=1}^d |u_i|^m\right)^{\frac{1}{m}}$,

[13]This can be seen as follows: $\mathbb{E}\|QC(\mathbf{u})\|^2 \le 2\mathbb{E}\|\mathbf{u} - QC(\mathbf{u})\|^2 + 2\|\mathbf{u}\|^2 \le 2(1 - \gamma)\|\mathbf{u}\|^2 + 2\|\mathbf{u}\|^2$.

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

# A    Supplementary Material of Section 2

Now we give a simple but important corollary, which allows us to apply different compression operators to different coordinates of a vector. For example, in the case of training neural networks, we can apply different compression operator to different layers.

**Corollary 3** (Piecewise Compression). *Let $C_i : \mathbb{R}^{d_i} \to \mathbb{R}^{d_i}$ for $i \in [L]$ denote possibly different compression operators with compression coefficients $\gamma_i$. Let $\mathbf{x} = [\mathbf{x}_1\, \mathbf{x}_2 \ldots \mathbf{x}_L]$, where $\mathbf{x}_i \in \mathbb{R}^{d_i}$ for all $i \in [L]$. Then $C(\mathbf{x}) := [C_1(\mathbf{x}_1)\, C_2(\mathbf{x}_2) \ldots C_L(\mathbf{x}_L)]$ is a compression operator with the compression coefficient being equal to $\gamma_{min} = \min_{i \in [L]} \gamma_i$.*

*Proof.* Fix an arbitrary $\mathbf{x} \in \mathbb{R}^d$.

$$
\begin{aligned}
\mathbb{E}_C \|\mathbf{x} - C(\mathbf{x})\|_2^2 &= \sum_{i=1}^{L} \mathbb{E}_{C_i} \|\mathbf{x}_i - C_i(\mathbf{x}_i)\|_2^2 \\
&\overset{(a)}{\leq} \sum_{i=1}^{L} (1 - \gamma_i) \|\mathbf{x}_i\|_2^2 \\
&\leq (1 - \gamma_{min}) \|\mathbf{x}\|_2^2
\end{aligned}
$$

Inequality (a) follows because each $C_i$ is a compression operator with the compression coefficient $\gamma_i$. $\qquad\square$

**Lemma** (Restating Lemma 1, Composing stochastic quantization and sparsification). *Let $Comp_k \in \{\text{Top}_k, \text{Rand}_k\}$. Let $Q_s : \mathbb{R}^d \to \mathbb{R}^d$ be a stochastic quantizer with parameter $s$ that satisfies Definition 1. Let $Q_s Comp_k : \mathbb{R}^d \to \mathbb{R}^d$ be defined as $Q_s Comp_k(\mathbf{x}) := Q_s(Comp_k(\mathbf{x}))$ for every $\mathbf{x} \in \mathbb{R}^d$. Then $\frac{Q_s Comp_k(\mathbf{x})}{(1+\beta_{k,s})}$ is a compression operator with the compression coefficient being equal to $\gamma = \frac{k}{d(1+\beta_{k,s})}$.*

*Proof.* Fix an arbitrary $\mathbf{x} \in \mathbb{R}^d$.

$$
\begin{aligned}
\mathbb{E}_{C,Q}[\|\mathbf{x} - \tfrac{Q_s Comp_k(\mathbf{x})}{(1+\beta_{k,s})}\|_2^2] &= \|\mathbf{x}\|_2^2 - 2\mathbb{E}_C\left[\left\langle \mathbf{x}, \mathbb{E}_Q\left[\tfrac{Q_s Comp_k(\mathbf{x})}{(1+\beta_{k,s})}\right]\right\rangle\right] + \mathbb{E}_{C,Q}\left[\tfrac{\|Q_s Comp_k(\mathbf{x})\|_2^2}{(1+\beta_{k,s})^2}\right] \\
&\overset{(a)}{=} \|\mathbf{x}\|_2^2 - \tfrac{2}{(1+\beta_{k,s})}\mathbb{E}_C\left[\langle \mathbf{x}, Comp_k(\mathbf{x})\rangle\right] \\
&\qquad + \tfrac{1}{(1+\beta_{k,s})^2}\mathbb{E}_{C,Q}\left[\|Q_s Comp_k(\mathbf{x})\|_2^2\right]
\end{aligned}
$$

In (a) we used $\mathbb{E}_Q[Q_s Comp_k(\mathbf{x})] = Comp_k(\mathbf{x})$, which follows from (i) of Definition 1. Observe that, for $Comp_k \in \{\text{Top}_k, \text{Rand}_k\}$, we have $\langle \mathbf{x}, Comp_k(\mathbf{x})\rangle = \|Comp_k(\mathbf{x})\|_2^2$. Continuing from above, we get

$$
\begin{aligned}
\mathbb{E}_{C,Q}[\|\mathbf{x} - \tfrac{Q_s Comp_k(\mathbf{x})}{(1+\beta_{k,s})}\|_2^2] &= \|\mathbf{x}\|_2^2 - \tfrac{2}{(1+\beta_{k,s})}\mathbb{E}_C\left[\|Comp_k(\mathbf{x})\|_2^2\right] \\
&\qquad + \tfrac{1}{(1+\beta_{k,s})^2}\mathbb{E}_{C,Q}\left[\|Q_s Comp_k(\mathbf{x})\|_2^2\right] \qquad (13)
\end{aligned}
$$

Observe that for any $Comp_k \in \{\text{Top}_k, \text{Rand}_k\}$, $Comp_k(\mathbf{x})$ is a length-$d$ vector, but only (at most) $k$ of its components are non-zero. This implies that, by treating $Comp_k(\mathbf{x})$ a length-$k$ vector whose entries correspond to the $k$ non-zero entries of $\mathbf{x}$, we can write $\mathbb{E}_Q[\|Q_s Comp_k(\mathbf{x})\|_2^2] \leq (1 + \beta_{k,s})\|Comp_k(\mathbf{x})\|_2^2$; see (ii) of Definition 1. Putting this back in (13), we get

$$
\begin{aligned}
\mathbb{E}_{C,Q}[\|\mathbf{x} - \tfrac{Q_s Comp_k(\mathbf{x})}{(1+\beta_{k,s})}\|_2^2] &\leq \|\mathbf{x}\|_2^2 - \tfrac{2}{1+\beta_{k,s}}\mathbb{E}_C\left[\|Comp_k(\mathbf{x})\|_2^2\right] \\
&\qquad + \tfrac{1}{(1+\beta_{k,s})}\mathbb{E}_C\left[\|Comp_k(\mathbf{x})\|_2^2\right] \\
&= \|\mathbf{x}\|_2^2 - \tfrac{1}{(1+\beta_{k,s})}\mathbb{E}_C\left[\|Comp_k(\mathbf{x})\|_2^2\right] \qquad (14)
\end{aligned}
$$

Using $\mathbb{E}_C[\|Comp_k(\mathbf{x})\|_2^2] \geq \frac{k}{d}\|\mathbf{x}\|_2^2$ (see (16) in Lemma 5) in (14) gives

$$
\begin{aligned}
\mathbb{E}_{C,Q}[\|\mathbf{x} - \tfrac{Q_s Comp_k(\mathbf{x})}{(1+\beta_{k,s})}\|_2^2] &\leq \|\mathbf{x}\|_2^2 - \tfrac{(k/d)\|\mathbf{x}\|_2^2}{(1+\beta_{k,s})} \\
&= \left[1 - \tfrac{k}{d(1+\beta_{k,s})}\right]\|\mathbf{x}\|_2^2.
\end{aligned}
$$

This completes the proof of Lemma 1.

□

**Lemma 5.** *Let* $Comp_k \in \{\mathrm{Top}_k, \mathrm{Rand}_k\}$. *For any* $\mathbf{x} \in \mathbb{R}^d$, *we have*

$$\mathbb{E}[\|Comp_k(\mathbf{x})\|_1^2] \geq \max\left\{\tfrac{k}{d}\|\mathbf{x}\|_2^2, \tfrac{k^2}{d^2}\|\mathbf{x}\|_1^2\right\} \tag{15}$$

$$\mathbb{E}[\|Comp_k(\mathbf{x})\|_2^2] \geq \tfrac{k}{d}\|\mathbf{x}\|_2^2. \tag{16}$$

*Proof.* Let $m \in \{1, 2\}$. Observe that for any $\mathbf{x} \in \mathbb{R}^d$, we have $\mathbb{E}[\|\mathrm{Top}_k(\mathbf{x})\|_m^2] = \|\mathrm{Top}_k(\mathbf{x})\|_m^2$ and that $\|\mathrm{Top}_k(\mathbf{x})\|_m^2 \geq \mathbb{E}[\|\mathrm{Rand}_k(\mathbf{x})\|_m^2]$. So, in order to prove the lemma, it suffices to show that $\mathbb{E}[\|\mathrm{Rand}_k(\mathbf{x})\|_m^2] \geq \tfrac{k}{d}\|\mathbf{x}\|_m^2$ holds for any $m \in \{1, 2\}$, and that $\mathbb{E}[\|\mathrm{Rand}_k(\mathbf{x})\|_1^2] \geq \tfrac{k^2}{d^2}\|\mathbf{x}\|_1^2$. Let $\Omega_k$ be the set of all the $k$-elements subsets of $[d]$.

$$
\begin{aligned}
\mathbb{E}[\|\mathrm{Rand}_k(\mathbf{x})\|_m^2] &= \sum_{\omega \in \Omega_k} \tfrac{1}{|\Omega_k|} \left(\sum_{i=1}^d |x_i|^m \cdot \mathbb{1}\{i \in \omega\}\right)^{2/m} \\
&\overset{(a)}{\geq} \sum_{\omega \in \Omega_k} \tfrac{1}{|\Omega_k|} \sum_{i=1}^d |x_i|^2 \cdot \mathbb{1}\{i \in \omega\} \\
&= \sum_{i=1}^d x_i^2 \cdot \tfrac{1}{|\Omega_k|} \sum_{\omega \in \Omega_k} \mathbb{1}\{i \in \omega\} \\
&= \sum_{i=1}^d x_i^2 \cdot \tfrac{1}{|\Omega_k|} \binom{d-1}{k-1} \\
&= \tfrac{k}{d}\|\mathbf{x}\|_2^2
\end{aligned}
$$

Note that (a) holds only for $m \in \{1, 2\}$, and it is equality for $m = 2$. Now we show that $\mathbb{E}[\|\mathrm{Rand}_k(\mathbf{x})\|_1^2] \geq \tfrac{k^2}{d^2}\|\mathbf{x}\|_1^2$.

$$
\begin{aligned}
\mathbb{E}[\|\mathrm{Rand}_k(\mathbf{x})\|_1^2] &\geq \left(\mathbb{E}[\|\mathrm{Rand}_k(\mathbf{x})\|_1]\right)^2 \\
&= \left(\sum_{\omega \in \Omega_k} \tfrac{1}{|\Omega_k|} \sum_{i=1}^d |x_i| \cdot \mathbb{1}\{i \in \omega\}\right)^2 \\
&= \left(\sum_{i=1}^d |x_i| \cdot \tfrac{1}{|\Omega_k|} \sum_{\omega \in \Omega_k} \mathbb{1}\{i \in \omega\}\right)^2 \\
&= \left(\sum_{i=1}^d |x_i| \cdot \tfrac{1}{|\Omega_k|} \binom{d-1}{k-1}\right)^2 \\
&= \tfrac{k^2}{d^2}\|\mathbf{x}\|_1^2
\end{aligned}
$$

□

**Lemma 6.** *For* $Comp_k \in \{\mathrm{Top}_k, \mathrm{Rand}_k\}$, $\frac{\|Comp_k(\mathbf{x})\|_m \, SignComp_k(\mathbf{x})}{k}$, *where* $\|\cdot\|_m$ *is the $m$-norm,*[12] *for any* $m \in \mathbb{Z}_+$ *is a compression operator with the compression coefficient* $\gamma_m$ *being equal to*

$$
\gamma_m = \begin{cases} \max\left\{\tfrac{1}{d}, \tfrac{k}{d}\left(\tfrac{\|Comp_k(\mathbf{x})\|_1}{\sqrt{d}\|Comp_k(\mathbf{x})\|_2}\right)^2\right\} & \text{if } m = 1, \\ \tfrac{k^{\frac{2}{m}-1}}{d} & \text{if } m \geq 2. \end{cases}
$$

**Remark 1.** *Note that this subsumes Lemma 2, which is for $m = 1$. Observe that for $m = 1$, depending on the value of $k$, either of the terms inside the max can be bigger than the other term. For example, if $k = 1$, then $\|Comp_k(\mathbf{x})\|_1 = \|Comp_k(\mathbf{x})\|_2$, which implies that the second term inside the max is equal to $1/d^2$, which is much smaller than the first term. On the other hand, if $k = d$ and the vector $\mathbf{x}$ is dense, then the second term may be much bigger than the first term.*

*Proof of Lemma 6.* Take an arbitrary $\mathbf{x} \in \mathbb{R}^d$.

$$\mathbb{E}_C \left\| \frac{\|Comp_k(\mathbf{x})\|_m \, SignComp_k(\mathbf{x})}{k} - \mathbf{x} \right\|_2^2$$

$$= \mathbb{E}_C \left[ \frac{\|Comp_k(\mathbf{x})\|_m^2}{k} - 2 \left\langle \frac{\|Comp_k(\mathbf{x})\|_m \, SignComp_k(\mathbf{x})}{k}, \mathbf{x} \right\rangle + \|\mathbf{x}\|_2^2 \right]$$

$$= \mathbb{E}_C \left[ \frac{\|Comp_k(\mathbf{x})\|_m^2}{k} - 2 \frac{\|Comp_k(\mathbf{x})\|_m \|Comp_k(\mathbf{x})\|_1}{k} + \|\mathbf{x}\|_2^2 \right]$$

$$\leq \|\mathbf{x}\|_2^2 - \frac{\mathbb{E}_C \|Comp_k(\mathbf{x})\|_m^2}{k} \tag{17}$$

In (17) we used the fact that $\| \cdot \|_1 \geq \| \cdot \|_m$ for every $m \geq 1$.

Case 1. When $m = 1$: Substituting $\mathbb{E}_C \|Comp_k(\mathbf{x})\|_1^2 \geq \max\left\{ \frac{k}{d} \|\mathbf{x}\|_2^2, \frac{k^2}{d^2} \|\mathbf{x}\|_1^2 \right\}$ (from (15)) in (17) gives

$$\mathbb{E}_C \left\| \frac{\|Comp_k(\mathbf{x})\|_1 \, SignComp_k(\mathbf{x})}{k} - \mathbf{x} \right\|_2^2 \leq \|\mathbf{x}\|_2^2 - \frac{1}{k} \max\left\{ \frac{k}{d} \|\mathbf{x}\|_2^2, \frac{k^2}{d^2} \|\mathbf{x}\|_1^2 \right\}$$

$$\leq \left[ 1 - \max\left\{ \frac{1}{d}, \frac{k}{d} \left( \frac{\|Comp_k(\mathbf{x})\|_1}{\sqrt{d}\|Comp_k(\mathbf{x})\|_2} \right)^2 \right\} \right] \|\mathbf{x}\|_2^2.$$

Case 2. When $m \geq 2$: Since $\|\mathbf{u}\|_p \leq k^{\frac{1}{p} - \frac{1}{q}} \|\mathbf{u}\|_q$ holds for every $\mathbf{u} \in \mathbb{R}^k$, whenever $p \leq q$, using this in (17) with $q = m$ and $p = 2$ gives

$$\mathbb{E}_C \left\| \frac{\|Comp_k(\mathbf{x})\|_m \, SignComp_k(\mathbf{x})}{k} - \mathbf{x} \right\|_2^2$$

$$\leq \|\mathbf{x}\|_2^2 - \frac{1}{k} k^{\frac{2}{m} - 1} \mathbb{E}_C[\|Comp_k(\mathbf{x})\|_2^2]$$

$$\leq \|\mathbf{x}\|_2^2 - \frac{1}{k} k^{\frac{2}{m} - 1} (k/d) \|\mathbf{x}\|_2^2 \quad \text{(By Lemma 5)}$$

$$= \left[ 1 - \frac{k^{\frac{2}{m} - 1}}{d} \right] \|\mathbf{x}\|_2^2. \tag{18}$$

This completes the proof of Lemma 6. $\qquad\square$

# B  Supplementary Material of Synchronous Qsparse-local-SGD from Section 3

## B.1  Additional Theorem

Here we give a complementary result for Theorem 1, which was for a fixed learning rate. As noted earlier, the following theorem hold when Algorithm 1 is run with any compression operator (including our composed operators).

**Theorem 5** (Convergence in the smooth (non-convex) case with decaying learning rate). *Let $f^{(r)}(\mathbf{x})$ be $L$-smooth for every $r \in [R]$. Let $QComp_k : \mathbb{R}^d \to \mathbb{R}^d$ be a compression operator whose compression coefficient is equal to $\gamma \in (0, 1]$. Let $\{\widehat{\mathbf{x}}_t^{(r)}\}_{t=0}^{T-1}$ be generated according to Algorithm 1 with $QComp_k$, for step sizes $\eta_t = \frac{\xi}{(a+t)}$ and $gap(\mathcal{I}_T) \leq H$, where $a > 1$ is such that, we have $a > \max\{\frac{4H}{\gamma}, 2\xi L, H\}$ and $C \geq \frac{4a\gamma(1-\gamma^2)}{a\gamma - 4H}$. Then the following holds.*

$$\mathbb{E}\|\nabla f(\mathbf{z}_T)\|^2 \leq \frac{\mathbb{E}f(\mathbf{x}_0) - f^*}{P_T} + \frac{L\xi^2}{(a-1)P_T} \left( \frac{\sum_{r=1}^R \sigma_r^2}{bR^2} \right) + \left( \frac{8C}{\gamma^2} + 8 \right) \frac{\xi^3 L^2 G^2 H^2}{2(a-1)^2 P_T} \tag{19}$$

*Here (i) $\delta_t := \frac{\eta_t}{4R}$; (ii) $P_T := \sum_{t=0}^{T-1} \sum_{r=1}^R \delta_t$, which is lower bounded as $P_T \geq \frac{\xi}{4} \ln\left( \frac{T+a-1}{a} \right)$; and (iii) $\mathbf{z}_T$ is a random variable which samples a previous parameter $\widehat{\mathbf{x}}_t^{(r)}$ with probability $\delta_t/P_T$.*

## B.2  Maintain Virtual Sequences

As outlined in Section 3, in order to prove our results, we define virtual sequences for every worker $r \in [R]$ and for all $t \geq 0$ as follows:

$$\widetilde{\mathbf{x}}_0^{(r)} := \widehat{\mathbf{x}}_0^{(r)} \qquad \text{and} \qquad \widetilde{\mathbf{x}}_{t+1}^{(r)} := \widetilde{\mathbf{x}}_t^{(r)} - \eta_t \nabla f_{i_t^{(r)}}\left(\widehat{\mathbf{x}}_t^{(r)}\right) \tag{20}$$

Define (i) $\mathbf{p}_t := \frac{1}{R}\sum_{r=1}^{R} \nabla f_{i_t^{(r)}}\left(\widehat{\mathbf{x}}_t^{(r)}\right)$, $\quad \overline{\mathbf{p}}_t := \mathbb{E}_{i_t}[\mathbf{p}_t] = \frac{1}{R}\sum_{r=1}^{R} \nabla f^{(r)}\left(\widehat{\mathbf{x}}_t^{(r)}\right)$;

and (ii) $\widetilde{\mathbf{x}}_{t+1} := \frac{1}{R}\sum_{r=1}^{R} \widetilde{\mathbf{x}}_{t+1}^{(r)} = \widetilde{\mathbf{x}}_t - \eta_t \mathbf{p}_t$, $\quad \widehat{\mathbf{x}}_t := \frac{1}{R}\sum_{r=1}^{R} \widehat{\mathbf{x}}_t^{(r)}$

## B.3  Bounding Error Compensation (Memory)

### B.3.1  Difference of the true and the virtual parameter vectors is the average memory

**Lemma 7** (Memory). *The memory is maintained so as to capture the distance between the true sequence and virtual sequence.*

$$\widehat{\mathbf{x}}_t - \widetilde{\mathbf{x}}_t = \frac{1}{R}\sum_{r=1}^{R} m_t^{(r)}. \tag{21}$$

*Proof.* Recall notation for an intermediate variable $\widehat{\mathbf{x}}_{t+\frac{1}{2}}^{(r)}$ in Algorithm 1. Now consider $\widehat{\mathbf{x}}_t - \widetilde{\mathbf{x}}_t = \frac{1}{R}\sum_{r=1}^{R} \widehat{\mathbf{x}}_t^{(r)} - \widetilde{\mathbf{x}}_t^{(r)}$. For the nearest $t_r + 1 \in \mathcal{I}_T$ such that $t_r + 1 \leq t$ and the nearest $t_r' + 1 \in \mathcal{I}_T$ such that $t_r' + 1 \leq t_r$

$$\begin{aligned}
\widehat{\mathbf{x}}_t - \widetilde{\mathbf{x}}_t &= \frac{1}{R}\sum_{r=1}^{R} \left(\widehat{\mathbf{x}}_{t_r+1}^{(r)} - \widetilde{\mathbf{x}}_{t_r+1}^{(r)}\right) \\
&= \frac{1}{R}\sum_{r=1}^{R} \left(\mathbf{x}_{t_r} - \frac{1}{R}\sum_{r=1}^{R} g_{t_r}^{(r)} - (\widetilde{\mathbf{x}}_{t_r'+1}^{(r)} - (\widehat{\mathbf{x}}_{t_r'+1}^{(r)} - \widehat{\mathbf{x}}_{t_r+\frac{1}{2}}^{(r)}))\right)
\end{aligned} \tag{22}$$

Here we used that $\widehat{\mathbf{x}}_{t_r'+1}^{(r)} - \widehat{\mathbf{x}}_{t_r+\frac{1}{2}}^{(r)} = \sum_{j=t_r'+1}^{t_r} \eta_j \nabla f_{i_j^{(r)}}\left(\widehat{\mathbf{x}}_j^{(r)}\right)$. Substituting $\widehat{\mathbf{x}}_{t_r'+1}^{(r)} = \mathbf{x}_{t_r'+1}$ we get

$$\begin{aligned}
\widehat{\mathbf{x}}_t - \widetilde{\mathbf{x}}_t &= \frac{1}{R}\sum_{r=1}^{R} \left(\mathbf{x}_{t_r} - \frac{1}{R}\sum_{r=1}^{R} g_{t_r}^{(r)} - (\widetilde{\mathbf{x}}_{t_r'+1}^{(r)} - (\mathbf{x}_{t_r'+1} - \widehat{\mathbf{x}}_{t_r+\frac{1}{2}}^{(r)}))\right) \\
&= \mathbf{x}_{t_r'+1} - \frac{1}{R}\sum_{r=1}^{R} g_{t_r}^{(r)} - (\widetilde{\mathbf{x}}_{t_r'+1} - (\mathbf{x}_{t_r'+1} - \widehat{\mathbf{x}}_{t_r+\frac{1}{2}})) \\
&= \widehat{\mathbf{x}}_{t_r'+1} - \widetilde{\mathbf{x}}_{t_r'+1} + (\mathbf{x}_{t_r'+1} - \widehat{\mathbf{x}}_{t_r+\frac{1}{2}}) - \frac{1}{R}\sum_{r=1}^{R} g_{t_r}^{(r)}
\end{aligned} \tag{23}$$

Now since $\mathbf{x}_{t_r'+1} = \mathbf{x}_{t_r}$ we have

$$\widehat{\mathbf{x}}_t - \widetilde{\mathbf{x}}_t = \widehat{\mathbf{x}}_{t_r'+1} - \widetilde{\mathbf{x}}_{t_r'+1} + (\mathbf{x}_{t_r} - \widehat{\mathbf{x}}_{t_r+\frac{1}{2}}) - \frac{1}{R}\sum_{r=1}^{R} g_{t_r}^{(r)} \tag{24}$$

On rolling out the expression in (24) we get

$$\begin{aligned}
\widehat{\mathbf{x}}_t - \widetilde{\mathbf{x}}_t &= \frac{1}{R}\sum_{r=1}^{R} \left[\sum_{\substack{j:j+1\in\mathcal{I}_T \\ j\leq t_r}} \left(\mathbf{x}_j^{(r)} - \widehat{\mathbf{x}}_{j+\frac{1}{2}}^{(r)} - g_j^{(r)}\right)\right] \\
&= \frac{1}{R}\sum_{r=1}^{R} m_{t_r+1}^{(r)} \\
&= \frac{1}{R}\sum_{r=1}^{R} m_t^{(r)}
\end{aligned} \tag{25}$$

Therefore $\widehat{\mathbf{x}}_t - \widetilde{\mathbf{x}}_t = \frac{1}{R}\sum_{r=1}^{R} m_t^{(r)}$ is the average memory. $\qquad\qquad\square$

### B.3.2  Memory contraction under the decaying learning rate

**Lemma** (Restating Lemma 4, Memory Contraction). *Let $\mathcal{I}_T \in [T]$ be a set of time instances in which the worker $r$ updates and synchronizes with the master. For $a > \frac{4H}{\gamma}$, $\eta_t = \frac{\xi}{a+t}$, $gap(\mathcal{I}_T) \le H$ and $t \in \mathbb{Z}^+$, there exists a $C \ge \frac{4a\gamma(1-\gamma^2)}{a\gamma - 4H}$ such that*

$$\mathbb{E}\|m_t^{(r)}\|^2 \le 4\frac{\eta_t^2}{\gamma^2}CH^2G^2. \tag{26}$$

*Proof.* Fix an arbitrary worker $r \in [R]$. In order to prove the lemma, we need to show that $\mathbb{E}\|m_t^{(r)}\|^2 \le 4\frac{\eta_t^2}{\gamma^2}CH^2G^2$ holds for every $t \in [T]$, where $C \ge \frac{4a\gamma(1-\gamma^2)}{a\gamma-4H}$. We show this separately for two cases, depending on whether or not $t \in \mathcal{I}_T$. First consider the case when $t \in \mathcal{I}_T$. Let $\mathcal{I}_T = \{t_{(1)}, t_{(2)}, \ldots, t_{(l)} = T\}$. Fix any $i = 1, 2, \ldots, l$ and consider $\mathbb{E}\|m_{t_{(i+1)}}^{(r)}\|^2$. Note that local memory $m_t^{(r)}$ at any worker $r$ and the global parameter vector $\mathbf{x}_t$ do not change in between the synchronization indices. We define $m_{t_{(0)}}^{(r)} := \mathbf{0}$ for every $r \in [R]$.

$$
\begin{aligned}
\mathbb{E}\|m_{t_{(i+1)}}^{(r)}\|^2 &= \mathbb{E}\|m_{t_{(i+1)}-1}^{(r)} + \mathbf{x}_{t_{(i+1)}-1} - \widehat{\mathbf{x}}_{t_{(i+1)}-\frac{1}{2}}^{(r)} - g_{t_{(i+1)}-1}^{(r)}\|^2 \\
&\overset{(a)}{\le} (1-\gamma)\mathbb{E}\|m_{t_{(i+1)}-1}^{(r)} + \mathbf{x}_{t_{(i+1)}-1} - \widehat{\mathbf{x}}_{t_{(i+1)}-\frac{1}{2}}^{(r)}\|^2 \\
&\overset{(b)}{=} (1-\gamma)\,\mathbb{E}\|m_{t_{(i)}}^{(r)} + \mathbf{x}_{t_{(i)}} - \widehat{\mathbf{x}}_{t_{(i+1)}-\frac{1}{2}}^{(r)}\|^2 \\
&\overset{(c)}{=} (1-\gamma)\,\mathbb{E}\|m_{t_{(i)}}^{(r)} + \widehat{\mathbf{x}}_{t_{(i)}}^{(r)} - \widehat{\mathbf{x}}_{t_{(i+1)}-\frac{1}{2}}^{(r)}\|^2
\end{aligned}
\tag{27}
$$

Here (a) is due to the compression property, (b) holds since the memory and master parameter remain unchanged between two rounds of synchronization, and in (c) we used that $\widehat{\mathbf{x}}_{t_{(i)}}^{(r)} = \mathbf{x}_{t_{(i)}}$, which holds for every $r$. Using the inequality $\|\mathbf{a} + \mathbf{b}\|^2 \le (1+\tau)\|\mathbf{a}\|^2 + (1+\frac{1}{\tau})\|\mathbf{b}\|^2$, which holds for every $\tau > 0$, in (27) gives (take any $p > 1$ in the following):

$$
\begin{aligned}
\mathbb{E}\|m_{t_{(i+1)}}^{(r)}\|^2 &\le (1-\gamma)\left[\left(1 + \frac{(p-1)\gamma}{p}\right)\mathbb{E}\|m_{t_{(i)}}^{(r)}\|^2 + \left(1 + \frac{p}{(p-1)\gamma}\right)\mathbb{E}\|\widehat{\mathbf{x}}_{t_{(i)}}^{(r)} - \widehat{\mathbf{x}}_{t_{(i+1)}-\frac{1}{2}}^{(r)}\|^2\right] \\
&\le \left(1 - \frac{\gamma}{p}\right)\mathbb{E}\|m_{t_{(i)}}^{(r)}\|^2 + \frac{(1-\gamma)(p\gamma+p)}{(p-1)\gamma}\mathbb{E}\|\widehat{\mathbf{x}}_{t_{(i)}}^{(r)} - \widehat{\mathbf{x}}_{t_{(i+1)}-\frac{1}{2}}^{(r)}\|^2 \\
&= \left(1 - \frac{\gamma}{p}\right)\mathbb{E}\|m_{t_{(i)}}^{(r)}\|^2 + \frac{p(1-\gamma^2)}{(p-1)\gamma}\mathbb{E}\|\widehat{\mathbf{x}}_{t_{(i)}}^{(r)} - \widehat{\mathbf{x}}_{t_{(i+1)}-\frac{1}{2}}^{(r)}\|^2 \\
&= \left(1 - \frac{\gamma}{p}\right)\mathbb{E}\|m_{t_{(i)}}^{(r)}\|^2 + \frac{p(1-\gamma^2)}{(p-1)\gamma}\mathbb{E}\|\sum_{j=t_{(i)}}^{t_{(i+1)}-1}\eta_j\nabla f_{i_j^{(r)}}\left(\widehat{\mathbf{x}}_j^{(r)}\right)\|^2 \\
&\le \left(1 - \frac{\gamma}{p}\right)\mathbb{E}\|m_{t_{(i)}}^{(r)}\|^2 + \frac{p(1-\gamma^2)}{(p-1)\gamma}\eta_{t_{(i)}}^2 H^2 G^2
\end{aligned}
\tag{28}
$$

In the last inequality (28) we used $\mathbb{E}\|\sum_{j=t_{(i)}}^{t_{(i+1)}-1}\eta_j\nabla f_{i_j^{(r)}}\left(\widehat{\mathbf{x}}_j^{(r)}\right)\|^2 \le \eta_{t_{(i)}}^2 H^2 G^2$, which can be seen as follows:

$$
\begin{aligned}
\mathbb{E}\|\sum_{j=t_{(i)}}^{t_{(i+1)}-1}\eta_j\nabla^{(r)}f_{(i_j)}\left(\widehat{\mathbf{x}}_j^{(r)}\right)\|^2 &= (t_{(i+1)}-t_{(i)})^2\mathbb{E}\|\frac{1}{(t_{(i+1)}-t_{(i)})}\sum_{j=t_{(i)}}^{t_{(i+1)}-1}\eta_j\nabla f_{i_j^{(r)}}\left(\widehat{\mathbf{x}}_j^{(r)}\right)\|^2 \\
&\overset{(a)}{\le} (t_{(i+1)}-t_{(i)})\sum_{j=t_{(i)}}^{t_{(i+1)}-1}\mathbb{E}\|\eta_j\nabla f_{i_j^{(r)}}\left(\widehat{\mathbf{x}}_j^{(r)}\right)\|^2 \\
&\overset{(b)}{\le} (t_{(i+1)}-t_{(i)})\eta_{t_{(i)}}^2\sum_{j=t_{(i)}}^{t_{(i+1)}-1}\mathbb{E}\|\nabla f_{i_j^{(r)}}\left(\widehat{\mathbf{x}}_j^{(r)}\right)\|^2
\end{aligned}
$$

$$\leq (t_{(i+1)} - t_{(i)})\eta^2_{t_{(i)}}(t_{(i+1)} - t_{(i)})G^2$$
$$\overset{(c)}{\leq} \eta^2_{t_{(i)}} H^2 G^2$$

Here (a) holds by Jensen's inequality, (b) holds since since $\eta_t \leq \eta_{t_{(i)}} \forall t \geq t_{(i)}$ and (c) holds because $(t_{(i+1)} - t_{(i)}) \leq H$. Define $\tilde{\eta}_t = \frac{1}{a+t}$ and $A = \xi^2 H^2 G^2$. Using this in (28) gives

$$\mathbb{E}\|m^{(r)}_{t_{(i+1)}}\|^2 \leq \left(1 - \frac{\gamma}{p}\right)\mathbb{E}\|m^{(r)}_{t_{(i)}}\|^2 + \frac{p(1-\gamma^2)}{(p-1)\gamma}\tilde{\eta}^2_{t_{(i)}}A. \tag{29}$$

We want to show that $\mathbb{E}\|m^{(r)}_{t_{(i)}}\|^2 \leq 4C\frac{\tilde{\eta}^2_{t_{(i)}}}{\gamma^2}A$ holds for every $i = 1, 2, \ldots$, where $C \geq \frac{4a\gamma(1-\gamma^2)}{a\gamma-4H}$.
In fact we prove a slightly stronger bound that $\mathbb{E}\|m^{(r)}_{t_{(i)}}\|^2 \leq C\frac{\tilde{\eta}^2_{t_{(i)}}}{\gamma^2}A$ holds for every $i = 1, 2, \ldots$.
We prove this using induction on $i$.

*Base case* ($i = 1$): Note that $m^{(r)}_{t_{(1)}-1} = m^{(r)}_0 = \mathbf{0}$. Consider the following:

$$\mathbb{E}\|m^{(r)}_{t_{(1)}}\|^2 = \mathbb{E}\|\mathbf{x}_{t_{(1)}-1} - \widehat{\mathbf{x}}_{t_{(1)}-\frac{1}{2}} - g^{(r)}_{t_{(1)}-1}\|^2$$
$$\leq (1-\gamma)\mathbb{E}\|\mathbf{x}_{t_{(1)}-1} - \widehat{\mathbf{x}}_{t_{(1)}-\frac{1}{2}}\|^2$$
$$\overset{(a)}{=} (1-\gamma)\mathbb{E}\|\widehat{\mathbf{x}}^{(r)}_0 - \widehat{\mathbf{x}}_{t_{(1)}-\frac{1}{2}}\|^2$$
$$= (1-\gamma)\mathbb{E}\|\sum_{j=0}^{t_{(1)}-1} \eta_j \nabla f_{i^{(r)}_j}\left(\widehat{\mathbf{x}}^{(r)}_j\right)\|^2$$
$$\leq (1-\gamma)\eta^2_0 H^2 G^2$$
$$= (1-\gamma)\tilde{\eta}^2_0 A$$

Here (a) holds since $\mathbf{x}_{t_{(1)}-1} = \mathbf{x}_0 = \widehat{\mathbf{x}}^{(r)}_0$. It is easy to verify that $(1-\gamma)\tilde{\eta}^2_0 A \leq \frac{4a\gamma(1-\gamma^2)}{a\gamma-4H}\frac{\tilde{\eta}^2_{t_{(1)}}}{\gamma^2}A$.
To show this, we use $\frac{\tilde{\eta}_0}{\tilde{\eta}_{t_{(1)}}} = \frac{a+t_{(1)}}{a} \leq \frac{a+H}{a} \leq 2$, where the first inequality follows from $t_{(1)} \leq H$
and the second inequality follows from $a \geq H$. Now, since $C \geq \frac{4a\gamma(1-\gamma^2)}{a\gamma-4H}$, it follows that
$\mathbb{E}\|m^{(r)}_{t_{(1)}}\|^2 \leq C\frac{\tilde{\eta}^2_{t_{(1)}}}{\gamma^2}A$.

*Inductive case:* Assume $\mathbb{E}\|m^{(r)}_{(i)}\|^2 \leq C\frac{\tilde{\eta}^2_{t_{(i)}}}{\gamma^2}A$ for some $i \in \mathbb{Z}^+$. We need to show that
$\mathbb{E}\|m^{(r)}_{(i+1)}\|^2 \leq C\frac{\tilde{\eta}^2_{t_{(i+1)}}}{\gamma^2}A$. Using the inductive hypothesis in (29), we get

$$\mathbb{E}\|m^{(r)}_{(i+1)}\|^2 \leq \left(1 - \frac{\gamma}{p}\right)C\frac{\tilde{\eta}^2_{t_{(i)}}}{\gamma^2}A + \frac{p(1-\gamma^2)}{(p-1)\gamma}\tilde{\eta}^2_{t_{(i)}}A$$
$$= C\frac{\tilde{\eta}^2_{t_{(i)}}}{\gamma^2}A\left(1 - \frac{\gamma}{p} + \frac{p(1-\gamma^2)}{p-1}\frac{\gamma}{C}\right)$$
$$= C\frac{\tilde{\eta}^2_{t_{(i)}}}{\gamma^2}A\left(1 - \frac{\gamma}{p}\left(1 - \frac{p^2(1-\gamma^2)}{(p-1)C}\right)\right) \tag{30}$$

**Claim 1.** *For any $p > 1$, if $\frac{\gamma}{p}\left(1 - \frac{p^2(1-\gamma^2)}{(p-1)C}\right) \geq \frac{2H}{a}$, then $\tilde{\eta}^2_{t_{(i)}}\left(1 - \frac{\gamma}{p}\left(1 - \frac{p^2(1-\gamma^2)}{(p-1)C}\right)\right) \leq \tilde{\eta}^2_{t_{(i+1)}}$ holds.*

*Proof.* Let $\frac{\gamma}{p}\left(1 - \frac{p^2(1-\gamma^2)}{(p-1)C}\right) = \frac{\beta}{a}$. Since $t_{(i+1)} \leq t_{(i)} + H$ (which implies that $\tilde{\eta}^2_{t_{(i)}+H} \leq \tilde{\eta}^2_{t_{(i+1)}}$),
it suffices to show that $\tilde{\eta}^2_{t_{(i)}}\left(1 - \frac{\beta}{a}\right) \leq \tilde{\eta}^2_{t_{(i)}+H}$ holds whenever $\beta \geq 2H$. For simplicity of
notation, let $t = t_{(i)}$. Note that $\tilde{\eta}^2_t\left(1 - \frac{\beta}{a}\right) = \frac{(a-\beta)}{a(a+t)^2}$. We show below that if $\beta > 2H$, then
$a(a + t)^2 \geq (a + t + H)^2(a - \beta)$. This proves our claim, because now we have $\frac{(a-\beta)}{a(a+t)^2} \leq$

$\frac{(a-\beta)}{(a+t+H)^2(a-\beta)} = \frac{1}{(a+t+H)^2} = \tilde{\eta}_{t+H}^2$. It only remains to show that $a(a+t)^2 \le (a+t+H)^2(a-\beta)$ holds if $\beta \ge 2H$.

$$
\begin{aligned}
(a+t+H)^2(a-\beta) &= \left((a+t)^2 + H^2 + 2H(a+t)\right)(a-\beta) \\
&= a(a+t)^2 + aH^2 + 2Ha^2 + 2Hat - \beta(a+t)^2 - \beta H^2 - 2H\beta(a+t) \\
&= a(a+t)^2 + a(H^2 + 2Ht - 2\beta t - 2H\beta) + a^2(2H - \beta) \\
&\quad - \beta t^2 - \beta H^2 - 2H\beta t \\
&\le a(a+t)^2.
\end{aligned}
$$

The last inequality holds whenever $\beta \ge 2H$. $\qquad\square$

Therefore we need $\frac{\gamma}{p}\left(1 - \frac{p^2(1-\gamma^2)}{(p-1)C}\right) \ge \frac{2H}{a}$, which is equivalent to requiring $C \ge \frac{\gamma a p^2(1-\gamma^2)}{(p-1)(a\gamma - 2pH)}$, where $a > \frac{2pH}{\gamma}$. Since this holds for every $p > 1$, by substituting $p = 2$, we get $C \ge \frac{4\gamma a(1-\gamma^2)}{(a\gamma - 4H)}$. This together with (30) and Claim 1 implies that if $C \ge \frac{4\gamma a(1-\gamma^2)}{(a\gamma - 4H)}$, where $a > 4H/\gamma$, then $\mathbb{E}\|m_{(i+1)}^{(r)}\|^2 \le C\frac{\tilde{\eta}_{t_{(i+1)}}^2}{\gamma^2}A$ holds. This proves our inductive step.

We have shown that $\mathbb{E}\|m_t^{(r)}\|^2 \le 4C\frac{\tilde{\eta}_t^2}{\gamma^2}A$ holds when $t \in \mathcal{I}_T$. It only remains to show that $\mathbb{E}\|m_t^{(r)}\|^2 \le 4C\frac{\tilde{\eta}_t^2}{\gamma^2}A$ also holds when $t \in [T] \setminus \mathcal{I}_T$. Let $i \in \mathbb{Z}_+$ be such that $t_{(i)} \le t < t_{(i+1)}$, which implies that $\tilde{\eta}_{t_{(i)}} \le 2\tilde{\eta}_t$. Since local memory does not change in between the synchronization indices, we have that $m_t^{(r)} = m_{t_{(i)}}^{(r)}$. Thus we have $\mathbb{E}\|m_t^{(r)}\|^2 = \mathbb{E}\|m_{t_{(i)}}^{(r)}\|^2 \le C\frac{\tilde{\eta}_{t_{(i)}}^2}{\gamma^2}A \le 4C\frac{\tilde{\eta}_t^2}{\gamma^2}A$. This concludes the proof of Lemma 4. $\qquad\square$

### B.3.3 Bounded memory under the fixed learning rate

**Lemma** (Restating Lemma 3, Bounded Memory). *Let $\mathcal{I}_T^{(r)} \in [T]$ be a set of time instances in which the worker $r$ updates and synchronizes with the master. For $\eta_t = \eta$, $gap(\mathcal{I}_T) \le H$ and $t \in \mathbb{Z}^+$ we have*

$$
\mathbb{E}\|m_t^{(r)}\|^2 \le 4\frac{\eta^2(1-\gamma^2)}{\gamma^2}H^2G^2 \tag{31}
$$

*Proof.* Observe that (28) holds irrespective of the learning rate schedule. In particular, using a fixed learning rate $\eta_t = \eta$ for every $t$ gives

$$
\mathbb{E}\|m_{t_{(i+1)}}^{(r)}\|^2 \le \left(1 - \frac{\gamma}{p}\right)\mathbb{E}\|m_{t_{(i)}}^{(r)}\|^2 + \frac{p(1-\gamma^2)}{(p-1)\gamma}\eta^2H^2G^2
$$

When rolled out we see that the memory is upper bounded by a geometric sum.

$$
\begin{aligned}
\mathbb{E}\|m_{t_{(i+1)}}^{(r)}\|^2 &\le \frac{p(1-\gamma^2)}{(p-1)\gamma}\eta^2H^2G^2\sum_{j=0}^{\infty}\left(1 - \frac{\gamma}{p}\right)^j \\
&\le \frac{p^2(1-\gamma^2)}{(p-1)}\frac{\eta^2}{\gamma^2}H^2G^2.
\end{aligned}
$$

Note that the last inequality holds for every $p > 1$, and is minimized when $p = 2$. By plugging $p = 2$, we get

$$
\mathbb{E}\|m_{t_{(i+1)}}^{(r)}\|^2 \le \frac{4(1-\gamma^2)\eta^2}{\gamma^2}H^2G^2.
$$

Since the RHS does not depend on $t$, it follows that $\mathbb{E}\|m_t^{(r)}\|^2 \le \frac{4(1-\gamma^2)\eta^2}{\gamma^2}H^2G^2$ holds for every $t \in [T]$. $\qquad\square$

### B.4 Sequence Deviation

#### B.4.1 Contracting deviation of local true sequences from the global true sequence under decaying learning rate

**Lemma 8** (Contracting Deviation of Local Sequences). *Similar to Lemma 3.3 in [31] we bound the deviation of the local sequences.*

$$\frac{1}{R}\sum_{r=1}^{R}\mathbb{E}\|\widehat{\mathbf{x}}_t - \widehat{\mathbf{x}}_t^{(r)}\|^2 \le 4\eta_t^2 G^2 H^2 \tag{32}$$

*Proof.* We need to upper-bound $\frac{1}{R}\sum_{r=1}^{R}\mathbb{E}\|\widehat{\mathbf{x}}_t - \widehat{\mathbf{x}}_t^{(r)}\|^2$. Note that for any $R$ vectors $\mathbf{u}_1, \ldots, \mathbf{u}_R$, if we let $\bar{\mathbf{u}} = \frac{1}{R}\sum_{i=1}^{r}\mathbf{u}_i$, then $\sum_{i=1}^{n}\|\mathbf{u}_i - \bar{\mathbf{u}}\|^2 \le \sum_{i=1}^{R}\|\mathbf{u}_i\|^2$. We use this in the first inequality below.

$$
\begin{aligned}
\frac{1}{R}\sum_{r=1}^{R}\mathbb{E}\|\widehat{\mathbf{x}}_t - \widehat{\mathbf{x}}_t^{(r)}\|^2 &= \frac{1}{R}\sum_{r=1}^{R}\mathbb{E}\|\widehat{\mathbf{x}}_t^{(r)} - \widehat{\mathbf{x}}_{t_r}^{(r)} - (\widehat{\mathbf{x}}_t - \widehat{\mathbf{x}}_{t_r}^{(r)})\|^2 \\
&\le \frac{1}{R}\sum_{r=1}^{R}\mathbb{E}\|\widehat{\mathbf{x}}_t^{(r)} - \widehat{\mathbf{x}}_{t_r}^{(r)}\|^2 \\
&\le \eta_{t_r}^2 G^2 H^2 \\
&\le 4\eta_t^2 G^2 H^2
\end{aligned}
\tag{33}
$$

The last inequality (33) uses $\eta_{t_r} \le 2\eta_{t_r+H} \le 2\eta_t$ and $t - t_r \le H$. $\qquad\square$

#### B.4.2 Bounded deviation of local true sequences from the global true sequence under fixed learning rate

**Lemma 9** (Bounded Deviation of Local Sequences). *With $\eta_t = \eta$ this follows from the analysis of Lemma 8*

$$\frac{1}{R}\sum_{r=1}^{R}\mathbb{E}\|\widehat{\mathbf{x}}_t - \widehat{\mathbf{x}}_t^{(r)}\|^2 \le \eta^2 G^2 H^2 \tag{34}$$

*Proof.* Similar to analysis in (33) we can show that $\frac{1}{R}\sum_{r=1}^{R}\mathbb{E}\|\widehat{\mathbf{x}}_t - \widehat{\mathbf{x}}_t^{(r)}\|^2 \le \eta^2 G^2 H^2$. $\qquad\square$

### B.5 Smooth and Non-Convex Objective

#### B.5.1 Proof of Theorem 1 – Fixed Learning Rate

*Proof.* Let $\mathbf{x}^*$ be the minimizer of $f(\mathbf{x})$, therefore we denote $f(\mathbf{x}^*)$ by $f^*$. For the purpose of reusing the proof later while proving Theorem 5, we start off with the decaying learning rate $\eta_t$ until (38) and then switch to the fixed learning rate $\eta$. Note that the proof remains the same until (38) irrespective of the learning rate schedule; in particular, we can take $\eta_t = \eta$ and the same proof holds until (38).

By the definition of $L$-smoothness, we have

$$
\begin{aligned}
f(\widetilde{\mathbf{x}}_{t+1}) - f(\widetilde{\mathbf{x}}_t) &\le \langle \nabla f(\widetilde{\mathbf{x}}_t), \widetilde{\mathbf{x}}_{t+1} - \widetilde{\mathbf{x}}_t \rangle + \frac{L}{2}\|\widetilde{\mathbf{x}}_{t+1} - \widetilde{\mathbf{x}}_t\|^2 \\
&= -\eta_t \langle \nabla f(\widetilde{\mathbf{x}}_t), \mathbf{p}_t \rangle + \frac{\eta_t^2 L}{2}\|\mathbf{p}_t\|^2 \\
&= -\eta_t \langle \nabla f(\widetilde{\mathbf{x}}_t), \mathbf{p}_t \rangle + \frac{\eta_t^2 L}{2}\|\mathbf{p}_t - \overline{\mathbf{p}}_t + \overline{\mathbf{p}}_t\|^2 \\
&\le -\eta_t \langle \nabla f(\widetilde{\mathbf{x}}_t), \mathbf{p}_t \rangle + \eta_t^2 L\|\mathbf{p}_t - \overline{\mathbf{p}}_t\|^2 + \eta_t^2 L\|\overline{\mathbf{p}}_t\|^2 \quad \text{(Using Jensen's Inequality)} \\
&= -\frac{\eta_t}{R}\sum_{r=1}^{R}\langle \nabla f(\widetilde{\mathbf{x}}_t), \nabla f_{i_t^{(r)}}(\widehat{\mathbf{x}}_t^{(r)}) \rangle + \eta_t^2 L\|\frac{1}{R}\sum_{r=1}^{R}\nabla f^{(r)}(\widehat{\mathbf{x}}_t^{(r)})\|^2 + \eta_t^2 L\|\mathbf{p}_t - \overline{\mathbf{p}}_t\|^2
\end{aligned}
$$

Define $i_t$ as the set of random sampling of the mini-batches at each worker $\{i_t^{(1)}, i_t^{(2)}, \ldots, i_t^{(R)}\}$. Taking expectation w.r.t. the sampling at time $t$ (conditioned on the past) and using the lipschitz

continuity of the gradients of local functions gives

$$\mathbb{E}_{i_t}[f(\widetilde{\mathbf{x}}_{t+1})] - f(\widetilde{\mathbf{x}}_t) \leq -\frac{\eta_t}{2}\left(\|\nabla f(\widetilde{\mathbf{x}}_t)\|^2 + \|\frac{1}{R}\sum_{r=1}^{R}\nabla f^{(r)}(\widehat{\mathbf{x}}_t^{(r)})\|^2 - \|\nabla f(\widetilde{\mathbf{x}}_t) - \frac{1}{R}\sum_{r=1}^{R}\nabla f^{(r)}(\widehat{\mathbf{x}}_t^{(r)})\|^2\right)$$

$$+ \eta_t^2 L\|\frac{1}{R}\sum_{r=1}^{R}\nabla f^{(r)}(\widehat{\mathbf{x}}_t^{(r)})\|^2 + \frac{\eta_t^2 L}{bR^2}\sum_{r=1}^{R}\sigma_r^2$$

$$\leq -\frac{\eta_t}{2R}\sum_{r=1}^{R}\left(\|\nabla f(\widetilde{\mathbf{x}}_t)\|^2 - L^2\|\widetilde{\mathbf{x}}_t - \widehat{\mathbf{x}}_t^{(r)}\|^2\right) + \frac{2\eta_t^2 L - \eta_t}{2}\|\frac{1}{R}\sum_{r=1}^{R}\nabla f^{(r)}(\widehat{\mathbf{x}}_t^{(r)})\|^2$$

$$+ \frac{\eta_t^2 L}{bR^2}\sum_{r=1}^{R}\sigma_r^2$$

$$= -\frac{\eta_t}{2R}\sum_{r=1}^{R}\left(\|\nabla f(\widetilde{\mathbf{x}}_t)\|^2 + L^2\|\widetilde{\mathbf{x}}_t - \widehat{\mathbf{x}}_t^{(r)}\|^2\right) + \frac{2\eta_t^2 L - \eta_t}{2R}\sum_{r=1}^{R}\|\nabla f(\widehat{\mathbf{x}}_t^{(r)})\|^2$$

$$+ \frac{\eta_t^2 L}{bR^2}\sum_{r=1}^{R}\sigma_r^2 + \frac{\eta_t L^2}{R}\sum_{r=1}^{R}\|\widetilde{\mathbf{x}}_t - \widehat{\mathbf{x}}_t^{(r)}\|^2. \tag{35}$$

We bound the first term in terms of $\|\nabla f(\widehat{\mathbf{x}}_t^{(r)})\|^2$ as follows:

$$\|\nabla f(\widehat{\mathbf{x}}_t^{(r)})\|^2 \leq 2\|\nabla f(\widehat{\mathbf{x}}_t^{(r)}) - \nabla f(\widetilde{\mathbf{x}}_t)\|^2 + 2\|\nabla f(\widetilde{\mathbf{x}}_t)\|^2$$

$$\leq 2L^2\|\widehat{\mathbf{x}}_t^{(r)} - \widetilde{\mathbf{x}}_t\|^2 + 2\|\nabla f(\widetilde{\mathbf{x}}_t)\|^2, \tag{36}$$

where the 2nd inequality follows from the smoothness ($L$-Lipschitz gradient) assumption. Using this and that $\eta_t \leq \frac{1}{2L}$ in (35) and rearranging terms give

$$\frac{\eta_t}{4R}\sum_{r=1}^{R}\|\nabla f(\widehat{\mathbf{x}}_t^{(r)})\|^2 \leq f(\widetilde{\mathbf{x}}_t) - \mathbb{E}_{(i_t)}[f(\widetilde{\mathbf{x}}_{t+1})] + \frac{\eta_t^2 L}{bR^2}\sum_{r=1}^{R}\sigma_r^2 + \frac{\eta_t L^2}{R}\sum_{r=1}^{R}\|\widetilde{\mathbf{x}}_t - \widehat{\mathbf{x}}_t^{(r)}\|^2 \tag{37}$$

Taking expectation w.r.t. to the entire process and using the inequality $\|\mathbf{u} + \mathbf{v}\|^2 \leq 2\|\mathbf{u}\|^2 + 2\|\mathbf{v}\|^2$ gives

$$\frac{\eta_t}{4R}\sum_{r=1}^{R}\mathbb{E}\|\nabla f(\widehat{\mathbf{x}}_t^{(r)})\|^2 \leq \mathbb{E}[f(\widetilde{\mathbf{x}}_t)] - \mathbb{E}[f(\widetilde{\mathbf{x}}_{t+1})] + \frac{\eta_t^2 L}{bR^2}\sum_{r=1}^{R}\sigma_r^2 + 2\eta_t L^2\mathbb{E}\|\widetilde{\mathbf{x}}_t - \widehat{\mathbf{x}}_t\|^2$$

$$+ 2\eta_t L^2 \frac{1}{R}\sum_{r=1}^{R}\mathbb{E}\|\widehat{\mathbf{x}}_t - \widehat{\mathbf{x}}_t^{(r)}\|^2 \tag{38}$$

Observe that (38) holds irrespective of the learning rate schedule. In particular, if we take a fixed learning rate $\eta_t = \eta \leq \frac{1}{2L}$ in (38), we get

$$\frac{\eta}{4R}\sum_{r=1}^{R}\mathbb{E}\|\nabla f(\widehat{\mathbf{x}}_t^{(r)})\|^2 \leq \mathbb{E}[f(\widetilde{\mathbf{x}}_t)] - \mathbb{E}[f(\widetilde{\mathbf{x}}_{t+1})] + \frac{\eta^2 L}{bR^2}\sum_{r=1}^{R}\sigma_r^2 + 2\eta L^2\mathbb{E}\|\widetilde{\mathbf{x}}_t - \widehat{\mathbf{x}}_t\|^2$$

$$+ 2\eta L^2 \frac{1}{R}\sum_{r=1}^{R}\mathbb{E}\|\widehat{\mathbf{x}}_t - \widehat{\mathbf{x}}_t^{(r)}\|^2 \tag{39}$$

Lemma 7 and Lemma 3 together imply $\mathbb{E}\|\widehat{\mathbf{x}}_t - \widetilde{\mathbf{x}}_t\|^2 \leq \frac{4\eta^2(1-\gamma^2)}{\gamma^2}G^2 H^2$. We also have from Lemma 9 that $\frac{1}{R}\sum_{r=1}^{R}\mathbb{E}\|\widetilde{\mathbf{x}}_t - \widehat{\mathbf{x}}_t^{(r)}\|^2 \leq \eta^2 G^2 H^2$. Substituting these in (39) gives

$$\frac{\eta}{4R}\sum_{r=1}^{R}\mathbb{E}\|\nabla f(\widehat{\mathbf{x}}_t^{(r)})\|^2 \leq \mathbb{E}[f(\widetilde{\mathbf{x}}_t)] - \mathbb{E}[f(\widetilde{\mathbf{x}}_{t+1})] + \frac{\eta^2 L}{bR^2}\sum_{r=1}^{R}\sigma_r^2 + 8\frac{\eta^3(1-\gamma^2)}{\gamma^2}L^2 G^2 H^2$$

$$+ 2\eta^3 L^2 G^2 H^2 \tag{40}$$

By taking a telescopic sum from $t = 0$ to $t = T - 1$, we get

$$\frac{1}{4RT} \sum_{t=0}^{T-1} \sum_{r=1}^{R} \mathbb{E}\|\nabla f(\widehat{\mathbf{x}}_t^{(r)})\|^2 \leq \frac{\mathbb{E}[f(\widetilde{\mathbf{x}}_0)] - f^*}{\eta T} + \frac{\eta L}{bR^2} \sum_{r=1}^{R} \sigma_r^2 + 8\frac{\eta^2(1-\gamma^2)}{\gamma^2} L^2 G^2 H^2$$

$$+ 2\eta^2 L^2 G^2 H^2 \qquad (41)$$

Take $\eta = \frac{\widehat{C}}{\sqrt{T}}$, where $\widehat{C}$ is a constant (that satisfies $\widehat{C} < \frac{\sqrt{T}}{2L}$). For example, we can take $\widehat{C} = \frac{1}{2L}$. This gives

$$\frac{1}{RT} \sum_{t=0}^{T-1} \sum_{r=1}^{R} \mathbb{E}\|\nabla f(\widehat{\mathbf{x}}_t^{(r)})\|^2 \leq \left( \frac{\mathbb{E}[f(\mathbf{x}_0)] - f^*}{\widehat{C}} + \frac{\widehat{C}L}{bR^2} \sum_{r=1}^{R} \sigma_r^2 \right) \frac{4}{\sqrt{T}} + 8\left(4\frac{(1-\gamma^2)}{\gamma^2} + 1\right) \frac{\widehat{C}^2 L^2 G^2 H^2}{T}.$$

$$(42)$$

Sample a parameter $\mathbf{z}_T$ from $\left\{\widehat{\mathbf{x}}_t^{(r)}\right\}$ for $r = 1, \ldots, R$ and $t = 0, 1, \ldots, T - 1$ with probability $\Pr[\mathbf{z}_T = \widehat{\mathbf{x}}_t^{(r)}] = \frac{1}{RT}$, which implies $\mathbb{E}\|\mathbf{z}_T\|^2 = \frac{1}{RT} \sum_{t=0}^{T-1} \sum_{r=1}^{R} \mathbb{E}\|\nabla f(\widehat{\mathbf{x}}_t^{(r)})\|^2$. Using this in (42) gives

$$\mathbb{E}\|\mathbf{z}_T\|^2 = \left( \frac{\mathbb{E}[f(\mathbf{x}_0)] - f^*}{\widehat{C}} + \frac{\widehat{C}L}{bR^2} \sum_{r=1}^{R} \sigma_r^2 \right) \frac{4}{\sqrt{T}} + 8\left(4\frac{(1-\gamma^2)}{\gamma^2} + 1\right) \frac{\widehat{C}^2 L^2 G^2 H^2}{T}.$$

This completes the proof of Theorem 1. $\qquad\qquad\qquad\qquad\qquad\qquad\qquad\qquad\qquad\qquad\qquad\quad\square$

### B.5.2  Proof of Theorem 5 – Decaying Learning Rate

*Proof.* Observe that we can use the proof of Theorem 1 exactly until (38), for $\eta_t \leq \frac{1}{2L}$ (which follows from our assumption that $a \geq 2\xi L$), which gives

$$\frac{\eta_t}{4R} \sum_{r=1}^{R} \mathbb{E}\|\nabla f(\widehat{\mathbf{x}}_t^{(r)})\|^2 \leq \mathbb{E}[f(\widetilde{\mathbf{x}}_t)] - \mathbb{E}[f(\widetilde{\mathbf{x}}_{t+1})] + \frac{\eta_t^2 L}{bR^2} \sum_{r=1}^{R} \sigma_r^2 + 2\eta_t L^2 \mathbb{E}\|\widetilde{\mathbf{x}}_t - \widehat{\mathbf{x}}_t\|^2$$

$$+ 2\eta_t L^2 \frac{1}{R} \sum_{r=1}^{R} \mathbb{E}\|\widehat{\mathbf{x}}_t - \widehat{\mathbf{x}}_t^{(r)}\|^2 \qquad (43)$$

We have from Lemma 8 that $\frac{1}{R} \sum_{r=1}^{R} \mathbb{E}\|\widehat{\mathbf{x}}_t - \widehat{\mathbf{x}}_t^{(r)}\|^2 \leq 4\eta_t^2 G^2 H^2$. Lemma 7 and Lemma 4 together imply that $\mathbb{E}\|\widehat{\mathbf{x}}_t - \widetilde{\mathbf{x}}_t\|^2 \leq \frac{1}{R} \sum_{r=1}^{R} \|m_t^{(r)}\|^2 \leq C\frac{4\eta_t^2}{\gamma^2} G^2 H^2$. Using these bounds in (43) gives

$$\frac{\eta_t}{4R} \sum_{r=1}^{R} \mathbb{E}\|\nabla f(\widehat{\mathbf{x}}_t^{(r)})\|^2 \leq \mathbb{E}[f(\widetilde{\mathbf{x}}_t)] - \mathbb{E}[f(\widetilde{\mathbf{x}}_{t+1})] + \frac{\eta_t^2 L}{bR^2} \sum_{r=1}^{R} \sigma_r^2 + \frac{8\eta_t^3}{\gamma^2} CL^2 G^2 H^2 + 8\eta_t^3 L^2 G^2 H^2$$

Taking a telescopic sum from $t = 0$ to $t = T - 1$ gives

$$\sum_{t=0}^{T-1} \frac{\eta_t}{4R} \sum_{r=1}^{R} \mathbb{E}\|\nabla f(\widehat{\mathbf{x}}_t^{(r)})\|^2 \leq \mathbb{E}[f(\mathbf{x}_0)] - f^* + \frac{L\sum_{r=1}^{R}\sigma_r^2}{bR^2} \sum_{t=0}^{T-1} \eta_t^2 + \left(\frac{8C}{\gamma^2} + 8\right) L^2 G^2 H^2 \sum_{t=0}^{T-1} \eta_t^3.$$

$$(44)$$

Let $\delta_t := \frac{\eta_t}{4R}$ and $P_T := \sum_{t=0}^{T-1} \sum_{r=1}^{R} \delta_t$. We show at the end of this proof that $P_T \geq \frac{\xi}{4} \ln\left(\frac{T+a-1}{a}\right)$, $\sum_{t=0}^{T-1} \eta_t^2 \leq \frac{\xi^2}{a-1}$, and that $\sum_{t=0}^{T-1} \eta_t^3 \leq \frac{\xi^3}{2(a-1)^2}$. Using these in (44) yields

$$\frac{1}{P_T} \sum_{t=0}^{T-1} \sum_{r=1}^{R} \delta_t \mathbb{E}\|\nabla f(\widehat{\mathbf{x}}_t^{(r)})\|^2 \leq \frac{\mathbb{E}f(\mathbf{x}_0) - f^*}{P_T} + \frac{L\xi^2}{bR^2(a-1)} \frac{\sum_{r=1}^{R}\sigma^2}{P_T}$$

$$+ \left(\frac{8C}{\gamma^2} + 8\right) L^2 G^2 H^2 \frac{\xi^3}{2P_T(a-1)^2} \qquad (45)$$

We therefore can show a weak convergence result, i.e.,

$$\min_{t \in \{0, \ldots, T-1\}, \, r \in [R]} \mathbb{E}\|\nabla f(\widehat{\mathbf{x}}_t^{(r)})\|^2 \xrightarrow{T \to \infty} 0. \qquad (46)$$

Sample a parameter $\mathbf{z}_T$ from $\left\{\widehat{\mathbf{x}}_t^{(r)}\right\}$ for $r = 1, \ldots, R$ and $t = 0, 1, \ldots, T-1$ with probability $\Pr[\mathbf{z}_T = \widehat{\mathbf{x}}_t^{(r)}] = \frac{\delta_t}{P_T}$. This gives $\mathbb{E}\|\nabla f(\mathbf{z}_T)\|^2 = \frac{1}{P_T} \sum_{t=0}^{T-1} \sum_{r=1}^{R} \delta_t \mathbb{E}\|\nabla f(\widehat{\mathbf{x}}_t^{(r)})\|^2$. We therefore have the following from (45)

$$\mathbb{E}\|\nabla f(\mathbf{z}_T)\|^2 \leq \frac{\mathbb{E}f(\mathbf{x}_0) - f^*}{P_T} + \frac{L\xi^2 \sum_{r=1}^{R} \sigma^2}{bR^2(a-1)P_T} + \left(\frac{8C}{\gamma^2} + 8\right)\frac{\xi^3 L^2 G^2 H^2}{2(a-1)^2 P_T}$$

Since $\min_{t \in \{0, \ldots, T-1\}, \, r \in [R]} \mathbb{E}\|\nabla f(\widehat{\mathbf{x}}_t^{(r)})\|^2$, we have a weak convergence result:

$$\min_{t \in \{0, \ldots, T-1\}, \, r \in [R]} \mathbb{E}\|\nabla f(\widehat{\mathbf{x}}_t^{(r)})\|^2 \xrightarrow{T \to \infty} 0.$$

Bounding the terms $P_T$, $\sum_{t=0}^{T-1} \eta_t^2$ and $\sum_{t=0}^{T-1} \eta_t^3$:

$$P_T = \frac{1}{4}\sum_{t=0}^{T-1} \eta_t \geq \frac{1}{4}\sum_{t=0}^{T-1} \eta_t \geq \frac{\xi}{4} \ln\left(\frac{T+a-1}{a}\right)$$

$$\sum_{t=0}^{T-1} \eta_t^2 \leq \xi^2 \left(\frac{1}{a-1} - \frac{1}{T+a-1}\right) = \frac{\xi^2 T}{(a-1)(T+a-1)} \leq \frac{\xi^2}{a-1}$$

$$\sum_{t=0}^{T-1} \eta_t^3 \leq \frac{\xi^3}{2}\left(\frac{1}{(a-1)^2} - \frac{1}{(T+a-1)^2}\right) \leq \frac{\xi^3}{2(a-1)^2}$$

This completes the proof of Theorem 5. $\qquad\qquad\square$

## B.6 Smooth and Strongly Convex Objective: Proof of Theorem 2 – Decaying Learning Rate

*Proof.* Let $\mathbf{x}^*$ be the minimizer of $f(\mathbf{x})$, therefore we have $\nabla f(\mathbf{x}^*) = 0$. We denote $f(\mathbf{x}^*)$ by $f^*$. By taking the average of the virtual sequences $\widetilde{\mathbf{x}}_{t+1}^{(r)} = \widetilde{\mathbf{x}}_t^{(r)} - \eta_t \nabla f_{i_t^{(r)}}\left(\widehat{\mathbf{x}}_t^{(r)}\right)$ for each worker $r \in [R]$ and defining $\mathbf{p}_t := \frac{1}{R}\sum_{r=1}^{R} \nabla f_{i_t^{(r)}}\left(\widehat{\mathbf{x}}_t^{(r)}\right)$, we get

$$\widetilde{\mathbf{x}}_{t+1} = \widetilde{\mathbf{x}}_t - \eta_t \mathbf{p}_t. \tag{47}$$

Define $i_t$ as the set of random sampling of the mini-batches at each worker $\{i_t^{(1)}, i_t^{(2)}, \ldots, i_t^{(R)}\}$ and let $\overline{\mathbf{p}}_t = \mathbb{E}_{i_t}[\mathbf{p}_t]$. From (47) we can get

$$\|\widetilde{\mathbf{x}}_{t+1} - \mathbf{x}^*\|^2 = \|\widetilde{\mathbf{x}}_t - \mathbf{x}^* - \eta_t \overline{\mathbf{p}}_t\|^2 + \eta_t^2 \|\mathbf{p}_t - \overline{\mathbf{p}}_t\|^2 - 2\eta_t \langle \widetilde{\mathbf{x}}_t - \mathbf{x}^* - \eta_t \overline{\mathbf{p}}_t, \mathbf{p}_t - \overline{\mathbf{p}}_t \rangle \tag{48}$$

Taking the expectation w.r.t. the sampling $i_t$ at time $t$ (conditioning on the past) and noting that last term in (48) becomes zero gives:

$$\mathbb{E}_{i_t}\|\widetilde{\mathbf{x}}_{t+1} - \mathbf{x}^*\|^2 = \|\widetilde{\mathbf{x}}_t - \mathbf{x}^* - \eta_t \overline{\mathbf{p}}_t\|^2 + \eta_t^2 \mathbb{E}_{i_t}\|\mathbf{p}_t - \overline{\mathbf{p}}_t\|^2 \tag{49}$$

It follows from the Jensen's inequality and independence that $\mathbb{E}_{i_t}\|\mathbf{p}_t - \overline{\mathbf{p}}_t\|^2 \leq \frac{\sum_{r=1}^{R} \sigma_r^2}{bR^2}$. This gives

$$\mathbb{E}_{i_t}\|\widetilde{\mathbf{x}}_{t+1} - \mathbf{x}^*\|^2 \leq \|\widetilde{\mathbf{x}}_t - \mathbf{x}^* - \eta_t \overline{\mathbf{p}}_t\|^2 + \eta_t^2 \frac{\sum_{r=1}^{R} \sigma_r^2}{bR^2}. \tag{50}$$

Now we bound the first term on the RHS.

**Lemma 10.** *If $\eta_t \leq \frac{1}{4L}$, then we have*

$$\|\widetilde{\mathbf{x}}_t - \mathbf{x}^* - \eta_t \overline{\mathbf{p}}_t\|^2 \leq \left(1 - \frac{\mu \eta_t}{2}\right)\|\widetilde{\mathbf{x}}_t - \mathbf{x}^*\|^2 - \frac{\eta_t \mu}{2L}(f(\widehat{\mathbf{x}}_t) - f^*)$$

$$+ \eta_t \left(\frac{3\mu}{2} + 3L\right)\|\widehat{\mathbf{x}}_t - \widetilde{\mathbf{x}}_t\|^2 + \frac{3\eta_t L}{R}\sum_{r=1}^{R}\|\widehat{\mathbf{x}}_t - \widehat{\mathbf{x}}_t^{(r)}\|^2 \tag{51}$$

*Proof.*

$$\|\widetilde{\mathbf{x}}_t - \mathbf{x}^* - \eta_t \overline{\mathbf{p}}_t\|^2 = \|\widetilde{\mathbf{x}}_t - \mathbf{x}^*\|^2 + \eta_t^2\|\overline{\mathbf{p}}_t\|^2 - 2\eta_t \langle \widetilde{\mathbf{x}}_t - \mathbf{x}^*, \overline{\mathbf{p}}_t \rangle \tag{52}$$

Using the definition of $\overline{\mathbf{p}}_t$ we have

$$\|\overline{\mathbf{p}}_t\|^2 = \|\tfrac{1}{R}\sum_{r=1}^{R}\left(\nabla f^{(r)}\left(\widehat{\mathbf{x}}_t^{(r)}\right) - \nabla f^{(r)}(\widetilde{\mathbf{x}}_t)\right) + \nabla f(\widetilde{\mathbf{x}}_t) - \nabla f(\mathbf{x}^*)\|^2$$

$$\leq \tfrac{1}{R}\sum_{r=1}^{R}2\|\nabla f^{(r)}\left(\widehat{\mathbf{x}}_t^{(r)}\right) - \nabla f^{(r)}(\widetilde{\mathbf{x}}_t)\|^2 + 2\|\nabla f(\widetilde{\mathbf{x}}_t) - \nabla f(\mathbf{x}^*)\|^2$$

$$\leq \tfrac{2L^2}{R}\sum_{r=1}^{R}\|\widehat{\mathbf{x}}_t^{(r)} - \widetilde{\mathbf{x}}_t\| + 2\|\nabla f(\widetilde{\mathbf{x}}_t) - \nabla f(\mathbf{x}^*)\|^2 \tag{53}$$

By the definition of smoothness, we have $\|\nabla f\left(\widetilde{\mathbf{x}}_t\right) - \nabla f\left(\mathbf{x}^*\right)\|^2 \leq 2L\left(f\left(\widetilde{\mathbf{x}}_t\right) - f(\mathbf{x}^*)\right)$, where $\nabla f(\mathbf{x}^*) = 0$. Substituting this in (53) gives

$$\eta_t^2\|\overline{\mathbf{p}}_t\|^2 \leq \tfrac{2\eta_t^2 L^2}{R}\sum_{r=1}^{R}\|\widehat{\mathbf{x}}_t^{(r)} - \widetilde{\mathbf{x}}_t\| + 4\eta_t^2 L\left(f\left(\widetilde{\mathbf{x}}_t\right) - f(\mathbf{x}^*)\right) \tag{54}$$

Now we bound the last term of (52). By definition, we have

$$-2\eta_t\left\langle\widetilde{\mathbf{x}}_t - \mathbf{x}^*, \overline{\mathbf{p}}_t\right\rangle = -2\tfrac{\eta_t}{R}\sum_{r=1}^{R}\left\langle\widehat{\mathbf{x}}_t^{(r)} - \mathbf{x}^*, \nabla f^{(r)}\left(\widehat{\mathbf{x}}_t^{(r)}\right)\right\rangle - 2\tfrac{\eta_t}{R}\sum_{r=1}^{R}\left\langle\widetilde{\mathbf{x}}_t - \widehat{\mathbf{x}}_t^{(r)}, \nabla f^{(r)}\left(\widehat{\mathbf{x}}_t^{(r)}\right)\right\rangle \tag{55}$$

For the first term on the RHS of (55), we can use strong convexity

$$-2\left\langle\widehat{\mathbf{x}}_t^{(r)} - \mathbf{x}^*, \nabla f^{(r)}\left(\widehat{\mathbf{x}}_t^{(r)}\right)\right\rangle \leq -2\left(f^{(r)}\left(\widehat{\mathbf{x}}_t^{(r)}\right) - f^{(r)}(\mathbf{x}^*)\right) - \mu\|\widehat{\mathbf{x}}_t^{(r)} - \mathbf{x}^*\|^2 \tag{56}$$

For the second term on the RHS of (55), we can use the following by smoothness.

$$-2\left\langle\widetilde{\mathbf{x}}_t - \widehat{\mathbf{x}}_t^{(r)}, \nabla f^{(r)}\left(\widehat{\mathbf{x}}_t^{(r)}\right)\right\rangle \leq L\|\widetilde{\mathbf{x}}_t - \widehat{\mathbf{x}}_t^{(r)}\|^2 + 2\left(f^{(r)}\left(\widehat{\mathbf{x}}_t^{(r)}\right) - f^{(r)}\left(\widetilde{\mathbf{x}}_t\right)\right) \tag{57}$$

Using (56)-(57) in (55) we get

$$-2\eta_t\left\langle\widetilde{\mathbf{x}}_t - \mathbf{x}^*, \overline{\mathbf{p}}_t\right\rangle \leq -\tfrac{2\eta_t}{R}\sum_{r=1}^{R}\left(f^{(r)}\left(\widetilde{\mathbf{x}}_t\right) - f^{(r)}(\mathbf{x}^*)\right) - \tfrac{\eta_t\mu}{R}\sum_{r=1}^{R}\|\widehat{\mathbf{x}}_t^{(r)} - \mathbf{x}^*\|^2 + \tfrac{L\eta_t}{R}\sum_{r=1}^{R}\|\widetilde{\mathbf{x}}_t - \widehat{\mathbf{x}}_t^{(r)}\|^2$$

$$= -2\eta_t\left(f\left(\widetilde{\mathbf{x}}_t\right) - f(\mathbf{x}^*)\right) - \tfrac{\eta_t\mu}{R}\sum_{r=1}^{R}\|\widehat{\mathbf{x}}_t^{(r)} - \mathbf{x}^*\|^2 + L\tfrac{\eta_t}{R}\sum_{r=1}^{R}\|\widetilde{\mathbf{x}}_t - \widehat{\mathbf{x}}_t^{(r)}\|^2 \tag{58}$$

Adding (54) and (58) and using $a \geq 32L/\mu$ which implies $\eta_t \leq 1/4L$ yields

$$\eta_t^2\|\overline{\mathbf{p}}_t\|^2 - 2\eta_t\left\langle\widetilde{\mathbf{x}}_t - \mathbf{x}^*, \overline{\mathbf{p}}_t\right\rangle \leq -2\eta_t(1 - 2\eta_t L)\left(f\left(\widetilde{\mathbf{x}}_t\right) - f^*\right) - \tfrac{\eta_t\mu}{R}\sum_{r=1}^{R}\|\widehat{\mathbf{x}}_t^{(r)} - \mathbf{x}^*\|^2$$

$$+ \tfrac{L\eta_t + 2\eta_t^2 L^2}{R}\sum_{r=1}^{R}\|\widetilde{\mathbf{x}}_t - \widehat{\mathbf{x}}_t^{(r)}\|^2$$

$$\leq -\eta_t\left(f\left(\widetilde{\mathbf{x}}_t\right) - f^*\right) - \eta_t\mu\|\widehat{\mathbf{x}}_t - \mathbf{x}^*\|^2$$

$$+ \tfrac{3L\eta_t}{R}\sum_{r=1}^{R}\left(\|\widetilde{\mathbf{x}}_t - \widehat{\mathbf{x}}_t\|^2 + \|\widehat{\mathbf{x}}_t - \widehat{\mathbf{x}}_t^{(r)}\|^2\right) \tag{59}$$

Since $\|\mathbf{x} + \mathbf{y}\|^2 \leq 2\|\mathbf{x}\|^2 + 2\|\mathbf{y}\|^2$, we have

$$-\|\widehat{\mathbf{x}}_t - \mathbf{x}^*\|^2 \leq \|\widehat{\mathbf{x}}_t - \widetilde{\mathbf{x}}_t\|^2 - \tfrac{1}{2}\|\widetilde{\mathbf{x}}_t - \mathbf{x}^*\|^2 \tag{60}$$

Using (60) in (59) and then substituting (59) in (52) gives

$$\|\widetilde{\mathbf{x}}_t - \mathbf{x}^* - \eta_t\overline{\mathbf{p}}_t\|^2 \leq \left(1 - \tfrac{\mu\eta_t}{2}\right)\|\widetilde{\mathbf{x}}_t - \mathbf{x}^*\|^2 - \eta_t\left(f\left(\widetilde{\mathbf{x}}_t\right) - f^*\right)$$

$$+ \eta_t\left(\mu + 3L\right)\|\widehat{\mathbf{x}}_t - \widetilde{\mathbf{x}}_t\|^2 + \tfrac{3L\eta_t}{R}\sum_{r=1}^{R}\|\widehat{\mathbf{x}}_t - \widehat{\mathbf{x}}_t^{(r)}\|^2 \tag{61}$$

Using strong convexity of $f$ we have

$$\|\widetilde{\mathbf{x}}_t - \mathbf{x}^* - \eta_t \overline{\mathbf{p}}_t\|^2 \le \left(1 - \tfrac{\mu\eta_t}{2}\right)\|\widetilde{\mathbf{x}}_t - \mathbf{x}^*\|^2 - \tfrac{\eta_t\mu}{2}\|\widetilde{\mathbf{x}}_t - \mathbf{x}^*\|^2$$
$$+ \eta_t(\mu + 3L)\|\widehat{\mathbf{x}}_t - \widetilde{\mathbf{x}}_t\|^2 + \tfrac{3L\eta_t}{R}\sum_{r=1}^{R}\|\widehat{\mathbf{x}}_t - \widehat{\mathbf{x}}_t^{(r)}\|^2 \tag{62}$$

Now use $-\|\widetilde{\mathbf{x}}_t - \mathbf{x}^*\|^2 \le \|\widetilde{\mathbf{x}}_t - \widehat{\mathbf{x}}_t\|^2 - \tfrac{1}{2}\|\widehat{\mathbf{x}}_t - \mathbf{x}^*\|^2$ We get

$$\|\widetilde{\mathbf{x}}_t - \mathbf{x}^* - \eta_t\overline{\mathbf{p}}_t\|^2 \le \left(1 - \tfrac{\mu\eta_t}{2}\right)\|\widetilde{\mathbf{x}}_t - \mathbf{x}^*\|^2 - \tfrac{\eta_t\mu}{4}\|\widehat{\mathbf{x}}_t - \mathbf{x}^*\|^2$$
$$+ \eta_t\left(\tfrac{3\mu}{2} + 3L\right)\|\widehat{\mathbf{x}}_t - \widetilde{\mathbf{x}}_t\|^2 + \tfrac{3L\eta_t}{R}\sum_{r=1}^{R}\|\widehat{\mathbf{x}}_t - \widehat{\mathbf{x}}_t^{(r)}\|^2$$
$$\le \left(1 - \tfrac{\mu\eta_t}{2}\right)\|\widetilde{\mathbf{x}}_t - \mathbf{x}^*\|^2 - \tfrac{\eta_t\mu}{2L}(f(\widehat{\mathbf{x}}_t) - f^*) \quad \text{(Using smoothness of } f(\mathbf{x}))$$
$$+ \eta_t\left(\tfrac{3\mu}{2} + 3L\right)\|\widehat{\mathbf{x}}_t - \widetilde{\mathbf{x}}_t\|^2 + \tfrac{3L\eta_t}{R}\sum_{r=1}^{R}\|\widehat{\mathbf{x}}_t - \widehat{\mathbf{x}}_t^{(r)}\|^2 \tag{63}$$

This completes the proof of Lemma 10. $\qquad\square$

Using (63) in (50) and then taking the expectation over the entire process gives

$$\mathbb{E}\|\widetilde{\mathbf{x}}_{t+1} - \mathbf{x}^*\|^2 \le \left(1 - \tfrac{\mu\eta_t}{2}\right)\mathbb{E}\|\widetilde{\mathbf{x}}_t - \mathbf{x}^*\|^2 - \tfrac{\eta_t\mu}{2L}(\mathbb{E}[f(\widehat{\mathbf{x}}_t)] - f^*)$$
$$+ \eta_t\left(\tfrac{3\mu}{2} + 3L\right)\mathbb{E}\|\widehat{\mathbf{x}}_t - \widetilde{\mathbf{x}}_t\|^2 + \frac{3\eta_t L}{R}\sum_{r=1}^{R}\mathbb{E}\|\widehat{\mathbf{x}}_t - \widehat{\mathbf{x}}_t^{(r)}\|^2 + \eta_t^2\frac{\sum_{r=1}^{R}\sigma_r^2}{bR^2} \tag{64}$$

From Lemma 8, we have $\frac{1}{R}\sum_{r=1}^{R}\mathbb{E}\|\widehat{\mathbf{x}}_t - \widehat{\mathbf{x}}_t^{(r)}\|^2 \le 4\eta_t^2 G^2 H^2$. Lemma 7 and Lemma 4 together imply that $\mathbb{E}\|\widehat{\mathbf{x}}_t - \widetilde{\mathbf{x}}_t\|^2 \le 4C\frac{\eta_t^2}{\gamma^2}H^2 G^2$. Substituting these back in (64) and letting $e_t = \mathbb{E}[f(\widehat{\mathbf{x}}_t) - f^*]$ gives

$$\mathbb{E}\|\widetilde{\mathbf{x}}_{t+1} - \mathbf{x}^*\|^2 \le \left(1 - \tfrac{\mu\eta_t}{2}\right)\mathbb{E}\|\widetilde{\mathbf{x}}_t - \mathbf{x}^*\|^2 - \tfrac{\mu\eta_t}{2L}e_t + \eta_t\left(\tfrac{3\mu}{2} + 3L\right)C\frac{4\eta_t^2}{\gamma^2}G^2 H^2$$
$$+ (3L\eta_t)4\eta_t^2 L G^2 H^2 + \eta_t^2\frac{\sum_{r=1}^{R}\sigma_r^2}{bR^2} \tag{65}$$

Now using $\eta_t \le 1/4L$ we have

$$\mathbb{E}\|\widetilde{\mathbf{x}}_{t+1} - \mathbf{x}^*\|^2 \le \left(1 - \tfrac{\mu\eta_t}{2}\right)\mathbb{E}\|\widetilde{\mathbf{x}}_t - \mathbf{x}^*\|^2 - \tfrac{\mu\eta_t}{2L}e_t + \eta_t\left(\tfrac{3\mu}{2} + 3L\right)C\frac{4\eta_t^2}{\gamma^2}G^2 H^2$$
$$+ (3\eta_t L)4\eta_t^2 L G^2 H^2 + \eta_t^2\frac{\sum_{r=1}^{R}\sigma_r^2}{bR^2} \tag{66}$$

Employing a slightly modified Lemma 3.3 from [30] with $A = \frac{\sum_{r=1}^{R}\sigma_r^2}{bR^2}$, $B = 4\left(\left(\tfrac{3\mu}{2} + 3L\right)\frac{CG^2 H^2}{\gamma^2} + 3L^2 G^2 H^2\right)$, and $a_t = \mathbb{E}\|\widetilde{\mathbf{x}}_t - \mathbf{x}^*\|^2$, we have

$$a_{t+1} \le \left(1 - \tfrac{\mu\eta_t}{2}\right)a_t - \tfrac{\mu\eta_t}{2L}e_t + \eta_t^2 A + \eta_t^3 B \tag{67}$$

For $\eta_t = \frac{8}{\mu(a+t)}$ and $w_t = (a+t)^2$, $S_T = \sum_{t=o}^{T-1} \ge \frac{T^3}{3}$ we have

$$\frac{\mu}{2LS_T}\sum_{t=0}^{T-1} w_t e_t \le \frac{\mu a^3}{8S_T}a_0 + \frac{4T(T+2a)}{\mu S_T}A + \frac{64T}{\mu^2 S_T}B \tag{68}$$

From convexity we can finally write

$$\mathbb{E}f(\overline{\mathbf{x}}_T) - f^* \le \frac{La^3}{4S_T}a_0 + \frac{8LT(T+2a)}{\mu^2 S_T}A + \frac{128LT}{\mu^3 S_T}B \tag{69}$$

Where $\overline{\mathbf{x}}_T := \frac{1}{S_T}\sum_{t=0}^{T-1}\left[w_t\left(\frac{1}{R}\sum_{r=1}^{R}\widehat{\mathbf{x}}_t^{(r)}\right)\right] = \frac{1}{S_T}\sum_{t=0}^{T-1} w_t\widehat{\mathbf{x}}_t$ $\qquad\square$

---

**Algorithm 2** Qsparse-local-SGD with asynchronous updates

---

1: Initialize $\mathbf{x}_0 = \bar{\bar{\mathbf{x}}}_0 = \mathbf{x}_0^{(r)} = \widehat{\mathbf{x}}_0^{(r)} = m_0^{(r)} = \mathbf{0}$, $\forall r \in [R]$. Suppose $\eta_t$ follows a certain learning rate schedule.
2: **for** $t = 0$ to $T - 1$ **do**
3:    **On Workers:**
4:    **for** $r = 1$ **to** $R$ **do**
5:       $\widehat{\mathbf{x}}_{t+\frac{1}{2}}^{(r)} \leftarrow \widehat{\mathbf{x}}_t^{(r)} - \eta_t \nabla f_{i_t^{(r)}}\left(\widehat{\mathbf{x}}_t^{(r)}\right)$; $i_t^{(r)}$ is a mini-batch of size $b$ uniformly in $\mathcal{D}_r$
6:       **if** $t + 1 \notin \mathcal{I}_T^{(r)}$ **then**
7:          $\mathbf{x}_{t+1}^{(r)} \leftarrow \mathbf{x}_t^{(r)}, m_{t+1}^{(r)} \leftarrow m_t^{(r)}$ and $\widehat{\mathbf{x}}_{t+1}^{(r)} \leftarrow \widehat{\mathbf{x}}_{t+\frac{1}{2}}^{(r)}$
8:       **else**
9:          $g_t^{(r)} \leftarrow Q\,Comp_k\left(m_t^{(r)} + \mathbf{x}_t^{(r)} - \widehat{\mathbf{x}}_{t+\frac{1}{2}}^{(r)}\right)$ and send $g_t^{(r)}$ to the master
10:         $m_{t+1}^{(r)} \leftarrow m_t^{(r)} + \mathbf{x}_t^{(r)} - \widehat{\mathbf{x}}_{t+\frac{1}{2}}^{(r)} - g_t^{(r)}$
11:          Receive $\bar{\bar{\mathbf{x}}}_{t+1}$ from the master and set $\mathbf{x}_{t+1}^{(r)} \leftarrow \bar{\bar{\mathbf{x}}}_{t+1}$ and $\widehat{\mathbf{x}}_{t+1}^{(r)} \leftarrow \bar{\bar{\mathbf{x}}}_{t+1}$
12:       **end if**
13:    **end for**
14:    **At Master:**
15:    **if** $t + 1 \notin \mathcal{I}_T^{(r)}$ for all $r \in [R]$ **then**
16:       $\bar{\bar{\mathbf{x}}}_{t+1} \leftarrow \bar{\bar{\mathbf{x}}}_t$
17:    **else**
18:       Let $\mathcal{S} \subseteq [R]$ be the set of all workers $r$ such that master receives $g_t^{(r)}$ from $r$.
19:       Compute $\bar{\bar{\mathbf{x}}}_{t+1} \leftarrow \bar{\bar{\mathbf{x}}}_t - \frac{1}{R}\sum_{r \in \mathcal{S}} g_t^{(r)}$ and broadcast $\bar{\bar{\mathbf{x}}}_{t+1}$ to all the workers in $\mathcal{S}$.
20:    **end if**
21: **end for**

---

# C   Supplementary Material of Asynchronous Qsparse-local-SGD from Section 4

Our algorithm for Asynchronous Qsparce-local-SGD is given below.

Below we give more precise statements of Theorem 3 and Theorem 4 from Section 4 (see Theorem 6 and Theorem 7, respectively), along with an additional theorem (see Theorem 8), which is a complementary result for Theorem 3, which was for a fixed learning rate. As noted earlier, the following theorem hold when Algorithm 2 is run with any compression operator (including our composed operators).

**Theorem 6** (Convergence in the smooth and non-convex case with fixed learning rate). *Under the same conditions as in Theorem 1 with* $gap(\mathcal{I}_T^{(r)}) \leq H$ *and* $C_1 = (\frac{8}{\gamma^2} - 6)(4 - 2\gamma)$, *if* $\{\widehat{x}_t^{(r)}\}_{t=0}^{T-1}$ *is generated according to Algorithm 2, the following holds.*

$$\mathbb{E}\|\nabla f(\mathbf{z}_T)\|^2 \leq \left(\frac{\mathbb{E}[f(\mathbf{x}_0)] - f^*}{\widehat{C}} + \widehat{C}L\left(\frac{\sum_{r=1}^R \sigma_r^2}{bR^2}\right)\right)\frac{4}{\sqrt{T}} + 8\left(12\frac{(1-\gamma^2)}{\gamma^2} + (2 + 8C_1 H^2)\right)\frac{\widehat{C}^2 L^2 G^2 H^2}{T}.$$

*Here* (i) $\mathbf{z}_T$ *is a random variable which samples a previous parameter* $\widehat{\mathbf{x}}_t^{(r)}$ *with probability* $1/RT$; *and* (ii) $\widehat{C}$ *is a constant such that* $\frac{\widehat{C}}{\sqrt{T}} \leq \frac{1}{2L}$.

**Corollary.** *Under the same conditions as in Theorem 1 with* $gap(\mathcal{I}_T^{(r)}) \leq H$, *if* $\{\widehat{\mathbf{x}}_t^{(r)}\}_{t=0}^{T-1}$ *is generated according to Algorithm 2, the following holds, where* $\mathbb{E}[f(\mathbf{x}_0)] - f^* \leq J^2$, $\sigma_{max} = \max_{r \in [R]} \sigma_r$, *and* $\widehat{C}^2 = bR(\mathbb{E}[f(\mathbf{x}_0)] - f^*)/\sigma_{max}^2$.

$$\mathbb{E}\|\nabla f(\mathbf{z}_T)\|^2 \leq \mathcal{O}\left(\frac{J\sigma_{max}}{\sqrt{bRT}}\right) + \mathcal{O}\left(\frac{J^2 bRG^2}{\sigma_{max}^2 \gamma^2 T}(H^2 + H^4)\right), \tag{70}$$

*where* $\mathbf{z}_T$ *is a random variable which samples a previous parameter* $\widehat{\mathbf{x}}_t^{(r)}$ *with probability* $1/RT$. *In order to ensure that the compression does not affect the dominating terms while converging at a rate of* $\mathcal{O}\left(1/\sqrt{bRT}\right)$, *we would require* $H = \mathcal{O}\left(\sqrt{\gamma}T^{1/8}/(bR)^{3/8}\right)$.

Theorem 6 provides non asymptotic guarantees where we also observe that the compression comes for "free". The corresponding asymptotic result has been omitted to Appendix C.

**Theorem 7** (Convergence in the smooth and strongly convex case with decaying learning rate). *Under the same conditions as in Theorem 2 with $gap(\mathcal{I}_T^{(r)}) \le H$, if $\{\widehat{x}_t^{(r)}\}_{t=0}^{T-1}$ is generated according to Algorithm 2, the following holds.*

$$\mathbb{E}[f(\overline{\mathbf{x}}_T)] - f^* \le \frac{La^3}{4S_T}\|\mathbf{x}_0 - \mathbf{x}^*\|^2 + \frac{8LT(T+2a)}{\mu^2 S_T}A + \frac{128LT}{\mu^3 S_T}D \tag{71}$$

*Here* (i) $C \ge \frac{4a\gamma(1-\gamma^2)}{a\gamma-4H}$, $C_1 = 192(4-2\gamma)\left(1+\frac{C}{\gamma^2}\right)$, $C_2 = 8(4-2\gamma)(1+\frac{C}{\gamma^2})$; (ii) $A = \frac{\sum_{r=1}^R \sigma_r^2}{bR^2}$, $D = \left(\frac{3\mu}{2}+3L\right)\left(\frac{12CG^2H^2}{\gamma^2}+C_1\eta_t^2 H^4 G^2\right)+24(1+C_2 H^2)LG^2 H^2$; *and* (iii) $\overline{\mathbf{x}}_T$, $S_T$ *are as defined in Theorem 2.*

**Corollary.** *Under the same conditions as in Theorem 2 with $gap(\mathcal{I}_T^{(r)}) \le H$, $a > \max\{\frac{4H}{\gamma}, 32\kappa, H\}$, $\sigma_{max} = \max_{r \in [R]} \sigma_r$, if $\{\widehat{\mathbf{x}}_t^{(r)}\}_{t=0}^{T-1}$ is generated according to Algorithm 2, the following holds:*

$$\mathbb{E}[f(\overline{\mathbf{x}}_T)] - f^* \le \mathcal{O}\left(\frac{G^2 H^3}{\mu^2\gamma^3 T^3}\right) + \mathcal{O}\left(\frac{\sigma_{max}^2}{\mu^2 bRT} + \frac{H\sigma_{max}^2}{\mu^2 bR\gamma T^2}\right) + \mathcal{O}\left(\frac{G^2}{\mu^3\gamma^2 T^2}(H^2+H^4)\right), \tag{72}$$

*where $\overline{\mathbf{x}}_T$, $S_T$ are as defined in Theorem 2. In order to ensure that the compression does not affect the dominating terms while converging at a rate of $\mathcal{O}\left(1/(bRT)\right)$, we would require $H = \mathcal{O}\left(\sqrt{\gamma}(T/(bR))^{1/4}\right)$.*

### C.1 Additional Theorem

**Theorem 8** (Convergence in the smooth and non-convex case with decaying learning rate). *Let $f^{(r)}(\mathbf{x})$ be L-smooth for every $r \in [R]$. Let $QComp_k : \mathbb{R}^d \to \mathbb{R}^d$ be a compression operator whose compression coefficient is equal to $\gamma \in (0,1]$. Let $\{\widehat{\mathbf{x}}_t^{(r)}\}_{t=0}^{T-1}$ be generated according to Algorithm 1 with $QComp_k$, for step sizes $\eta_t = \frac{\xi}{(a+t)}$, $gap(\mathcal{I}_T^r) \le H$ for $r \in [R]$, where $a > 1$ is such that, we have $a > \max\{\frac{4H}{\gamma}, 2\xi L, H\}$, $C \ge \frac{4a\gamma(1-\gamma^2)}{a\gamma-4H}$. Then for $C' = (4-2\gamma)(1+\frac{C}{\gamma^2})$ the following holds.*

$$\mathbb{E}\|\nabla f(\mathbf{z}_T)\|^2 \le \frac{\mathbb{E}f(\mathbf{x}_0)-f^*}{P_T} + \frac{L\xi^2}{(a-1)P_T}\left(\frac{\sum_{r=1}^R \sigma_r^2}{bR^2}\right) + \left(16 + \frac{24C}{\gamma^2} + 200C'H^2\right)\frac{\xi^3 L^2 G^2 H^2}{2(a-1)^2 P_T} \tag{73}$$

*Here (i) $\delta_t := \frac{\eta_t}{4R}$; (ii) $P_T := \sum_{t=0}^{T-1}\sum_{r=1}^R \delta_t$, which is lower bounded as $P_T \ge \frac{\xi}{4}\ln\left(\frac{T+a-1}{a}\right)$; and (iii) $\mathbf{z}_T$ is a random variable which samples a previous parameter $\widehat{\mathbf{x}}_t^{(r)}$ with probability $\delta_t/P_T$.*

### C.2 Maintain Virtual Sequences

As noted earlier in Section 3 and also in Appendix B, in order to prove our results in the asynchronous setting, we define virtual sequences for every worker $r \in [R]$ and for all $t \ge 0$ as follows:

$$\widetilde{\mathbf{x}}_0^{(r)} := \widehat{\mathbf{x}}_0^{(r)} \qquad\qquad \widetilde{\mathbf{x}}_{t+1}^{(r)} := \widetilde{\mathbf{x}}_t^{(r)} - \eta_t \nabla f_{i_t^{(r)}}\left(\widehat{\mathbf{x}}_t^{(r)}\right)$$

Define

1. $\widetilde{\mathbf{x}}_{t+1} := \frac{1}{R}\sum_{r=1}^R \widetilde{\mathbf{x}}_{t+1}^{(r)} = \widetilde{\mathbf{x}}_t - \frac{\eta_t}{R}\sum_{r=1}^R \nabla f_{i_t^{(r)}}\left(\widehat{\mathbf{x}}_t^{(r)}\right)$

2. $\mathbf{p}_t := \frac{1}{R}\sum_{r=1}^R \nabla f_{i_t^{(r)}}\left(\widehat{\mathbf{x}}_t^{(r)}\right)$

3. $\overline{\mathbf{p}}_t := \mathbb{E}_{(i_t)}[\mathbf{p}_t] = \frac{1}{R}\sum_{r=1}^R \nabla f^{(r)}\left(\widehat{\mathbf{x}}_t^{(r)}\right)$

4. $\widehat{\mathbf{x}}_t = \frac{1}{R}\sum_{r=1}^R \widehat{\mathbf{x}}_t^{(r)}$

5. $\mathcal{I}_T^{(r)} = \{t_{(i)}^{(r)} : i \in \mathbb{Z}^+, t_{(i)}^{(r)} \in [T], |t_{(i)}^{(r)} - t_{(j)}^{(r)}| \le H, \forall |i-j| \le 1\}$

### C.3 Contracting local sequence deviation under decaying learning rate

**Lemma 11** (Contracting Local Sequence Deviation). *For $\widehat{\mathbf{x}}_t, \widehat{\mathbf{x}}_t^{(r)}$ generated according to Algorithm 2 and $gap(\mathcal{I}_T^{(r)}) \le H$ the following holds*

$$\frac{1}{R}\sum_{r=1}^R \mathbb{E}\|\widehat{\mathbf{x}}_t - \widehat{\mathbf{x}}_t^{(r)}\|^2 \le 8(1+C''H^2)\eta_t^2 G^2 H^2 \tag{74}$$

*Here $C'' = 8B(1+\frac{C}{\gamma^2})$ where B is from $\mathbb{E}_{Q,C}\|QComp_k(\mathbf{x})\|^2 \le B\|\mathbf{x}\|^2$ and $C \ge \frac{4a\gamma(1-\gamma^2)}{a\gamma-4H}$*

*Proof.* Fix a time $t$ and consider any worker $r \in [R]$. Let $t_r \in \mathcal{I}_T^{(r)}$ denote the last synchronization step until time $t$ for the $r$'th worker. Define $t_0' := \min_{r \in [R]} t_r$. We need to upper-bound $\frac{1}{R} \sum_{r=1}^{R} \mathbb{E}\|\widehat{\mathbf{x}}_t - \widehat{\mathbf{x}}_t^{(r)}\|^2$. Note that for any $R$ vectors $\mathbf{u}_1, \ldots, \mathbf{u}_R$, if we let $\bar{\mathbf{u}} = \frac{1}{R} \sum_{i=1}^{r} \mathbf{u}_i$, then $\sum_{i=1}^{n} \|\mathbf{u}_i - \bar{\mathbf{u}}\|^2 \le \sum_{i=1}^{R} \|\mathbf{u}_i\|^2$. We use this in the first inequality below.

$$
\begin{aligned}
\frac{1}{R} \sum_{r=1}^{R} \mathbb{E}\|\widehat{\mathbf{x}}_t - \widehat{\mathbf{x}}_t^{(r)}\|^2 &= \frac{1}{R} \sum_{r=1}^{R} \mathbb{E}\|\widehat{\mathbf{x}}_t^{(r)} - \bar{\bar{\mathbf{x}}}_{t_0'} - (\widehat{\mathbf{x}}_t - \bar{\bar{\mathbf{x}}}_{t_0'})\|^2 \\
&\le \frac{1}{R} \sum_{r=1}^{R} \mathbb{E}\|\widehat{\mathbf{x}}_t^{(r)} - \bar{\bar{\mathbf{x}}}_{t_0'}\|^2 \\
&\le \frac{2}{R} \sum_{r=1}^{R} \mathbb{E}\|\widehat{\mathbf{x}}_t^{(r)} - \widehat{\mathbf{x}}_{t_r}^{(r)}\|^2 + \frac{2}{R} \sum_{r=1}^{R} \mathbb{E}\|\widehat{\mathbf{x}}_{t_r}^{(r)} - \bar{\bar{\mathbf{x}}}_{t_0'}\|^2
\end{aligned} \tag{75}
$$

We bound both the terms separately. For the first term:

$$
\begin{aligned}
\mathbb{E}\|\widehat{\mathbf{x}}_t^{(r)} - \widehat{\mathbf{x}}_{t_r}^{(r)}\|^2 &= \mathbb{E}\| \sum_{j=t_r}^{t-1} \eta_j \nabla f_{i_j^{(r)}} \left( \widehat{\mathbf{x}}_j^{(r)} \right) \|^2 \\
&\le (t - t_r) \sum_{j=t_r}^{t-1} \mathbb{E}\|\eta_j \nabla f_{i_j^{(r)}} \left( \widehat{\mathbf{x}}_j^{(r)} \right) \|^2 \\
&\le (t - t_r)^2 \eta_{t_r}^2 G^2 \\
&\le 4\eta_t^2 H^2 G^2.
\end{aligned} \tag{76}
$$

The last inequality (76) uses $\eta_{t_r} \le 2\eta_{t_r + H} \le 2\eta_t$ and $t - t_r \le H$. To bound the second term of (75), note that we have

$$
\bar{\bar{\mathbf{x}}}_{t_r}^{(r)} = \bar{\bar{\mathbf{x}}}_{t_0'} - \frac{1}{R} \sum_{s=1}^{R} \sum_{j=t_0'}^{t_r - 1} \mathbb{1}\{j + 1 \in \mathcal{I}_T^{(s)}\} g_j^{(s)}. \tag{77}
$$

Note that $\widehat{\mathbf{x}}_{t_r}^{(r)} = \bar{\bar{\mathbf{x}}}_{t_r}^{(r)}$, because at synchronization steps, the local parameter vector becomes equal to the global parameter vector. Using this, the Jensen's inequality, and that $\|\mathbb{1}\{j + 1 \in \mathcal{I}_T^{(s)}\} g_j^{(s)}\|^2 \le \|g_j^{(s)}\|^2$, we can upper-bound (77) as

$$
\mathbb{E}\|\widehat{\mathbf{x}}_{t_r}^{(r)} - \bar{\bar{\mathbf{x}}}_{t_0'}\|^2 \le \frac{(t_r - t_0')}{R} \sum_{s=1}^{R} \sum_{j=t_0'}^{t_r} \mathbb{E}\|g_j^{(s)}\|^2 \tag{78}
$$

Now we bound $\mathbb{E}\|g_j^{(s)}\|^2$ for any $j \in \{t_0', \ldots, t_r\}$ and $s \in [R]$: Since $\mathbb{E}\|QC(\mathbf{u})\|^2 \le B\|\mathbf{u}\|^2$ holds for every $\mathbf{u}$, where $B = (4 - 2\gamma)$,[13] we have for any $s \in [R]$ that

$$
\mathbb{E}\|g_j^{(s)}\|^2 \le B\mathbb{E}\|m_j^{(s)} + \mathbf{x}_j^{(s)} - \widehat{\mathbf{x}}_{j+\frac{1}{2}}^{(s)}\|^2 \tag{79}
$$

$$
\le 2B\mathbb{E}\|m_j^{(s)}\|^2 + 2B\mathbb{E}\|\mathbf{x}_j^{(s)} - \widehat{\mathbf{x}}_{j+\frac{1}{2}}^{(s)}\|^2 \tag{80}
$$

Observe that the proof of Lemma 4 does not depend on the synchrony of the network; it only uses the fact that $gap(\mathcal{I}_T^{(s)}) \le H$ for any worker $s \in [R]$. Therefore, we can directly use Lemma 4 to bound the first term in (76) as $\mathbb{E}\|m_j^{(s)}\|^2 \le 4C\frac{\eta_j^2}{\gamma^2} H^2 G^2$. In order to bound the second term of (76), note that $\mathbf{x}_j^{(s)} = \widehat{\mathbf{x}}_{t_s}^{(s)}$, which implies that $\|\mathbf{x}_j^{(s)} - \widehat{\mathbf{x}}_{j+\frac{1}{2}}^{(s)}\|^2 = \| \sum_{l=t_s}^{j} \eta_l \nabla f_{i_l^{(s)}} \left( \widehat{\mathbf{x}}_l^{(s)} \right) \|^2$. Taking expectation yields $\mathbb{E}\|\mathbf{x}_j^{(s)} - \widehat{\mathbf{x}}_{j+\frac{1}{2}}^{(s)}\|^2 \le 4\eta_{t_s}^2 H^2 G^2 \le 4\eta_{t_0'}^2 H^2 G^2$, where in the second inequality we used $t_0' \le t_s$, which implies $\eta_{t_s} \le \eta_{t_0'}$. Using these in (80) gives

$$
\mathbb{E}\|g_j^{(s)}\|^2 \le 8B \left( 1 + \frac{C}{\gamma^2} \right) \eta_{t_0'}^2 H^2 G^2. \tag{81}
$$

Since $t'_0 \leq t \leq t'_0 + H$, we have $\eta_{t'_0} \leq 2\eta_{t'_0 + H} \leq 2\eta_t$. Putting the bound on $\mathbb{E}\|g_j^{(s)}\|^2$ (after substituting $\eta_{n'_0} \leq 2\eta_t$ in (81)) in (78) gives

$$\mathbb{E}\|\widehat{\mathbf{x}}_{t_r}^{(r)} - \bar{\bar{\mathbf{x}}}_{t'_0}\|^2 \leq 32B\left(1 + \tfrac{C}{\gamma^2}\right)\eta_t^2 H^4 G^2. \tag{82}$$

Putting this and the bound from (76) back in (75) gives

$$\frac{1}{R}\sum_{r=1}^{R}\mathbb{E}\|\widehat{\mathbf{x}}_t - \widehat{\mathbf{x}}_t^{(r)}\|^2 \leq 8\eta_t^2 H^2 G^2 + 64B\left(1 + \tfrac{C}{\gamma^2}\right)\eta_t^2 H^4 G^2$$

$$\leq 8\left[1 + 8BH^2\left(1 + \tfrac{C}{\gamma^2}\right)\right]\eta_t^2 H^2 G^2.$$

This completes the proof of Lemma 11. $\qquad\qquad\qquad\qquad\qquad\qquad\qquad\qquad\square$

## C.4 Bounded local sequence deviation under fixed learning rate

**Lemma 12** (Bounded Local Sequence Deviation). *For $\widehat{\mathbf{x}}_t, \widehat{\mathbf{x}}_t^{(r)}$ generated according to Algorithm 2 with $\eta_t = \eta$ the following holds*

$$\frac{1}{R}\sum_{r=1}^{R}\mathbb{E}\|\widehat{\mathbf{x}}_t - \widehat{\mathbf{x}}_t^{(r)}\|^2 \leq (2 + H^2 C')\eta^2 G^2 H^2 \tag{83}$$

*Here $C' = (\tfrac{16}{\gamma^2} - 12)B$ where $B$ is from $\mathbb{E}_{Q,C}\|QComp_k(\mathbf{x})\|^2 \leq B\|\mathbf{x}\|^2$.*

*Proof.* From (79) and (80) and using the fact that for a given $QC$ operator, we show that $\mathbb{E}\|QC(\mathbf{u})\|^2 \leq B\|\mathbf{u}\|^2$ holds for every $\mathbf{u}$

$$\mathbb{E}\|g_j^{(s)}\|^2 \leq 2B\mathbb{E}\|m_j^{(s)}\|^2 + 2B\eta^2 H^2 G^2$$

$$\leq 8B\frac{(1-\gamma^2)\eta^2}{\gamma^2}H^2 G^2 + 2\eta^2 BH^2 G^2$$

$$= 2B\left(\tfrac{4}{\gamma^2} - 3\right)\eta^2 H^2 G^2 \tag{84}$$

For a fixed learning rate $\eta$, using (84) and following similar analysis as in (76) we can bound the first term in (75) as follows

$$\mathbb{E}\|\widehat{\mathbf{x}}_t^{(r)} - \widehat{\mathbf{x}}_{t_r}^{(r)}\|^2 \leq \eta^2 H^2 G^2 \tag{85}$$

Similarly as in (77)-(81) we can bound the second term in (75) as follows

$$\mathbb{E}\|\widehat{\mathbf{x}}_{t_r}^{(r)} - \bar{\bar{\mathbf{x}}}_{t'_0}\|^2 \leq 2B\left(\tfrac{4}{\gamma^2} - 3\right)\eta^2 H^4 G^2 \tag{86}$$

Using (85) and (86) in (75) we can show that

$$\frac{1}{R}\sum_{r=1}^{R}\mathbb{E}\|\widehat{\mathbf{x}}_t - \widehat{\mathbf{x}}_t^{(r)}\|^2 \leq \left[2 + 4BH^2\left(\tfrac{4}{\gamma^2} - 3\right)\right]\eta^2 H^2 G^2 \tag{87}$$

$$\qquad\qquad\qquad\qquad\qquad\qquad\qquad\qquad\qquad\qquad\square$$

## C.5 Contracting distance between virtual and true sequence under decaying learning rate

**Lemma 13** (Contracting distance between Virtual and True Sequence). *Let $\mathcal{I}_T^{(r)} \in [T]$ be a set of time instances in which the worker $r$ updates and synchronizes with the master. For $a > \tfrac{4H}{\gamma}$, $\eta_t = \tfrac{\xi}{a+t}$, $gap(\mathcal{I}_T^{(r)} \leq H)$ and $t \in \mathbb{Z}^+$, there exists a $C \geq \tfrac{4a\gamma(1-\gamma^2)}{a\gamma - 4H}$ such that*

$$\mathbb{E}\|\widehat{\mathbf{x}}_t - \widetilde{\mathbf{x}}_t\|^2 \leq C'\eta_t^2 H^4 G^2 + 12C\frac{\eta_t^2}{\gamma^2}G^2 H^2 \tag{88}$$

*Here $C' = 192B\left(1 + \tfrac{C}{\gamma^2}\right)$ where $B$ is from $\mathbb{E}_{Q,C}\|QComp_k(\mathbf{x})\|^2 \leq B\|\mathbf{x}\|^2$.*

*Proof.* Fix a time $t$ and consider any worker $r \in [R]$. Let $t_r \in \mathcal{I}_T^{(r)}$ denote the last synchronization step until time $t$ for the $r$'th worker. Define $t_0' := \min_{r \in [R]} t_r$. We want to bound $\mathbb{E}\|\widehat{\mathbf{x}}_t - \widetilde{\mathbf{x}}_t\|^2$. Note that in the synchronous case, we have shown in Lemma 7 that $\widehat{\mathbf{x}}_t - \widehat{\mathbf{x}}_t = \frac{1}{R}\sum_{r=1}^R m_t^{(r)}$. This does not hold in the asynchronous setting, which makes upper-bounding $\mathbb{E}\|\widehat{\mathbf{x}}_t - \widetilde{\mathbf{x}}_t\|^2$ a bit more involved. By definition $\widehat{\mathbf{x}}_t - \widetilde{\mathbf{x}}_t = \frac{1}{R}\sum_{r=1}^R \left(\widehat{\mathbf{x}}_t^{(r)} - \widetilde{\mathbf{x}}_t^{(r)}\right)$. By the definition of virtual sequences and the update rule for $\widehat{\mathbf{x}}_t^{(r)}$, we also have $\widehat{\mathbf{x}}_t - \widetilde{\mathbf{x}}_t = \frac{1}{R}\sum_{r=1}^R \left(\widehat{\mathbf{x}}_{t_r}^{(r)} - \widetilde{\mathbf{x}}_{t_r}^{(r)}\right)$. This can be written as

$$\widehat{\mathbf{x}}_t - \widetilde{\mathbf{x}}_t = \left[\frac{1}{R}\sum_{r=1}^R \widehat{\mathbf{x}}_{t_r}^{(r)} - \bar{\bar{\mathbf{x}}}_{t_0'}\right] + \left[\bar{\bar{\mathbf{x}}}_{t_0'} - \bar{\bar{\mathbf{x}}}_t\right] + \left[\bar{\bar{\mathbf{x}}}_t - \frac{1}{R}\sum_{r=1}^R \widetilde{\mathbf{x}}_{t_r}^{(r)}\right] \tag{89}$$

Applying Jensen's inequality and taking expectation gives

$$\mathbb{E}\|\widehat{\mathbf{x}}_t - \widetilde{\mathbf{x}}_t\|^2 \le \left[\frac{3}{R}\sum_{r=1}^R \mathbb{E}\|\widehat{\mathbf{x}}_{t_r}^{(r)} - \bar{\bar{\mathbf{x}}}_{t_0'}\|^2\right] + \left[3\mathbb{E}\|\bar{\bar{\mathbf{x}}}_{t_0'} - \bar{\bar{\mathbf{x}}}_t\|^2\right] + \left[3\mathbb{E}\|\bar{\bar{\mathbf{x}}}_t - \frac{1}{R}\sum_{r=1}^R \widetilde{\mathbf{x}}_{t_r}^{(r)}\|^2\right] \tag{90}$$

We bound each of the three terms of (90) separately. We have upper-bounded the first term earlier in (82), which is

$$\mathbb{E}\|\widehat{\mathbf{x}}_{t_r}^{(r)} - \bar{\bar{\mathbf{x}}}_{t_0'}\|^2 \le 32B\left(1 + \frac{C}{\gamma^2}\right)\eta_t^2 H^4 G^2. \tag{91}$$

To bound the second term of (90), note that

$$\bar{\bar{\mathbf{x}}}_t = \bar{\bar{\mathbf{x}}}_0 - \frac{1}{R}\sum_{r=1}^R \sum_{j=0}^{t_r-1} \mathbb{1}\{j+1 \in \mathcal{I}_T^{(r)}\} g_j^{(r)} \tag{92}$$

$$= \bar{\bar{\mathbf{x}}}_{t_0'} - \frac{1}{R}\sum_{r=1}^R \sum_{j=t_0'}^{t_r-1} \mathbb{1}\{j+1 \in \mathcal{I}_T^{(r)}\} g_j^{(r)} \tag{93}$$

By applying Jensen's inequality, using $\|\mathbb{1}\{j+1 \in \mathcal{I}_T^{(r)}\} g_j^{(r)}\|^2 \le \|g_j^{(r)}\|^2$, and taking expectation, we can upper-bound (93) as

$$\mathbb{E}\|\bar{\bar{\mathbf{x}}}_{t_0'} - \bar{\bar{\mathbf{x}}}_t\|^2 \le \frac{(t_r - t_0')}{R}\sum_{r=1}^R \sum_{j=t_0'}^{t_r} \mathbb{E}\|g_j^{(r)}\|^2$$

Using the bound on $\mathbb{E}\|g_j^{(r)}\|^2$'s from (82) gives

$$\mathbb{E}\|\bar{\bar{\mathbf{x}}}_{t_0'} - \bar{\bar{\mathbf{x}}}_t\|^2 \le 32B\left(1 + \frac{C}{\gamma^2}\right)\eta_t^2 H^4 G^2. \tag{94}$$

To bound the last term of (90), note that

$$\widetilde{\mathbf{x}}_{t_r}^{(r)} = \bar{\bar{\mathbf{x}}}_0 - \sum_{j=0}^{t_r-1} \eta_j \nabla f_{i_j^{(r)}}\left(\widehat{\mathbf{x}}_j^{(r)}\right) \tag{95}$$

From (92) and (95), we can write

$$\bar{\bar{\mathbf{x}}}_t - \frac{1}{R}\sum_{r=1}^R \widetilde{\mathbf{x}}_{t_r}^{(r)} = \frac{1}{R}\sum_{r=1}^R \left[\sum_{j=0}^{t_r-1} \eta_j \nabla^{(r)} f_{(i_j)}\left(\widehat{\mathbf{x}}_j^{(r)}\right) - \sum_{j=0}^{t_r-1} \mathbb{1}\{j+1 \in \mathcal{I}_T^{(r)}\} g_j^{(r)}\right] \tag{96}$$

Let $t_r^{(1)}$ and $t_r^{(2)}$ be two consecutive synchronization steps in $\mathcal{I}_T^{(r)}$. Then, by the update rule of $\widehat{\mathbf{x}}_t^{(r)}$, we have $\widehat{\mathbf{x}}_{t_r^{(1)}}^{(r)} - \widehat{\mathbf{x}}_{t_r^{(2)}-\frac{1}{2}}^{(r)} = \sum_{j=t_r^{(1)}}^{t_r^{(2)}-1} \nabla f_{i_j^{(r)}}\left(\widehat{\mathbf{x}}_j^{(r)}\right)$. Since $\mathbf{x}_{t_r^{(1)}}^{(r)} = \widehat{\mathbf{x}}_{t_r^{(1)}}^{(r)}$ and the workers do not modify their local $\mathbf{x}_t^{(r)}$'s in between the synchronization steps, we have $\mathbf{x}_{t_r^{(2)}-1}^{(r)} = \mathbf{x}_{t_r^{(1)}}^{(r)} = \widehat{\mathbf{x}}_{t_r^{(1)}}^{(r)}$. Therefore, we can write

$$\mathbf{x}_{t_r^{(2)}-1}^{(r)} - \widehat{\mathbf{x}}_{t_r^{(2)}-\frac{1}{2}}^{(r)} = \sum_{j=t_r^{(1)}}^{t_r^{(2)}-1} \nabla f_{i_j^{(r)}}\left(\widehat{\mathbf{x}}_j^{(r)}\right). \tag{97}$$

Using (97) for every consecutive synchronization steps, we can equivalently write (96) as

$$\bar{\bar{\mathbf{x}}}_t - \frac{1}{R}\sum_{r=1}^{R}\widetilde{\mathbf{x}}_{t_r}^{(r)} = \frac{1}{R}\sum_{r=1}^{R}\left[\sum_{\substack{j:j+1\in\mathcal{I}_T^{(r)}\\j\leq t_r-1}}\left(\mathbf{x}_j^{(r)} - \widehat{\mathbf{x}}_{j+\frac{1}{2}}^{(r)} - g_j^{(r)}\right)\right]$$

$$= \frac{1}{R}\sum_{r=1}^{R}m_{t_r}^{(r)}$$

$$= \frac{1}{R}\sum_{r=1}^{R}m_t^{(r)} \qquad (98)$$

In the last inequality, we used the fact that the workers do not update their local memory in between the synchronization steps. For the reasons given in the proof of Lemma 11, we can directly apply Lemma 4 to bound the local memories and obtain $\mathbb{E}\|\frac{1}{R}\sum_{r=1}^{R}m_t^{(r)}\|^2 \leq \frac{1}{R}\sum_{r=1}^{R}\mathbb{E}\|m_t^{(r)}\|^2 \leq 4C\frac{\eta_t^2}{\gamma^2}G^2H^2$. This implies

$$\mathbb{E}\|\bar{\bar{\mathbf{x}}}_t - \frac{1}{R}\sum_{r=1}^{R}\widetilde{\mathbf{x}}_{t_r}^{(r)}\|^2 \leq 4C\frac{\eta_t^2}{\gamma^2}G^2H^2. \qquad (99)$$

Putting the bounds from (91), (94), and (99) in (90) gives

$$\mathbb{E}\|\widehat{\mathbf{x}}_t - \widetilde{\mathbf{x}}_t\|^2 \leq 192B\left(1+\frac{C}{\gamma^2}\right)\eta_t^2H^4G^2 + 12C\frac{\eta_t^2}{\gamma^2}G^2H^2$$

This completes the proof of Lemma 13. □

## C.6 Bounded distance between virtual and true sequence under fixed learning rate

**Lemma 14** (Bounded Distance between Virtual and True Sequence). *Let $\mathcal{I}_T^{(r)} \in [T]$ be a set of time instances in which the worker $r$ updates and synchronizes with the master. For $\eta_t = \eta$, $gap(\mathcal{I}_T^{(r)} \leq H)$ and $t \in \mathbb{Z}^+$ we have*

$$\mathbb{E}\|\widehat{\mathbf{x}}_t - \widetilde{\mathbf{x}}_t\|^2 \leq 6C'\eta^2H^4G^2 + \frac{12\eta^2(1-\gamma^2)}{\gamma^2}G^2H^2 \qquad (100)$$

*Here $C' = B\left(\frac{8}{\gamma^2}-6\right)$ where $B$ is from $\mathbb{E}_{Q,C}\|QComp_k(\mathbf{x})\|^2 \leq B\|\mathbf{x}\|^2$.*

*Proof.* For a constant learning rate the first term in (90) has been bounded earlier in (86). Following similar steps as in (93) we would have

$$\mathbb{E}\|\bar{\bar{\mathbf{x}}}_{t_0'} - \bar{\bar{\mathbf{x}}}_t\|^2 \leq 2B\left(\frac{4}{\gamma^2}-3\right)\eta^2H^4G^2 \qquad (101)$$

Finally using (86),(98), Lemma 3 and (101) in (90) we have

$$\mathbb{E}\|\widehat{\mathbf{x}}_t - \widetilde{\mathbf{x}}_t\|^2 \leq 12B\left(\frac{4}{\gamma^2}-3\right)\eta^2H^4G^2 + \frac{12\eta^2(1-\gamma^2)}{\gamma^2}G^2H^2 \qquad (102)$$

□

# D  Further Experiments

## D.1  Experiments for Convex Objective

The experiments in Figure 3-4 and in Figure 2 are in a synchronous distributed setting with 15 worker nodes, each processing a mini-batch size of 8 samples per iteration using the *MNIST* [19] handwritten digits dataset. The corresponding experiments for the asynchronous operation (as in Algorithm 2) are shown in Figure 5.

### D.1.1 Model Architecture

Define the softmax function as

$$h_{\mathbf{x},z}\left(a^{(i)}\right) = \frac{\exp\left(\mathbf{x}_j^T a^{(i)} + z^{(i)}\right)}{\sum_{l=1}^{L} \exp\left(\mathbf{x}_l^T a^{(i)} + z^{(l)}\right)}.$$

Our experiments are all for the softmax regression with a standard $\ell_2$ regularizer. The cost function is

$$-\frac{1}{n}\left(\sum_{i=1}^{n}\sum_{j=1}^{L} \mathbb{1}\{b^{(i)} = j\} \log h_{\mathbf{x},z}\left(a^{(i)}\right)\right) + \frac{\lambda}{2}\|\mathbf{x}\|^2,$$

where $a^{(i)} \in \mathbb{R}^d$, $b^{(i)} \in [L]$ are the data points, which can belong to one of the $L$ classes, and $\mathbf{x}_j \in \mathbb{R}^d$ for every $j \in [L]$, are columns of the parameter structured as follows

$$\mathbf{x} = [\mathbf{x}_1 \quad \mathbf{x}_2 \quad \dots \quad \mathbf{x}_L], \quad \mathbf{x}_j \in \mathbb{R}^d, \ \forall j \in [L],$$

and $z^{(i)}$ for every $i \in [L]$ are the biases to be learnt corresponding to every class. We set $\lambda$ to be $1/n$.

### D.1.2 Parameter Selection and Learning Rates

We use our composed operator $SignTop_k$ (from Lemma 6) for compression, and denote the resulting SGD algorithm by SignTopK. The schemes with which we compare SignTopK are EF-SIGNSGD [15], TopK-SGD [4, 30], and local SGD [31]. The learning rate used for training is of the form $\frac{c}{\lambda(a+t)}$, where (i) $\lambda$ is the regularization parameter; (ii) $c$ is set with a careful hyperparameter sweep; (iii) $w_t = (a+t)^2$ as in Theorem 2, where $a$ is set as $\frac{dH}{k}$ with $d$ being the dimension of the gradient vector (7850 for *MNIST*); (iv) $k = 40$ is the sparsity; (v) $H$ is the synchronization period; (vi) $t$ is the iteration index; (vii) $b = 8$ is the batch size; and (viii) $R = 15$ is the number of workers.

### D.1.3 Results

**(a)** Training loss vs epochs     **(b)** Training loss vs $\log_2$ of communication budget     **(c)** top-1 accuracy [18] for schemes in Figure 3a

**Figure 3** Figure 3a-3c demonstrate the gains in performance achieved by our $Qsparse$ operators in a convex setting.

In Figure 3a, we observe that the composition of a quantizer with a sparsifier has very little effect on the rate of convergence as compared to when the techniques are used individually. From Figure 3b and 3c, we see that our composed operator achieves gains in communicated bits by a factor of 6-8 times over the state-of-the-art.

Figure 4a, demonstrates the effect of incorporating local iterations in SignTopK, and we see that the rate of convergence is not significantly affected as we go from 1 to 8 local iterations. Furthermore, observe that for a fixed number of local iterations h, SignTopK_hL maintains the same rates as vanilla SGD or TopK-SGD. In doing so, it is able to achieve gains in communicated bits as seen in Figure 4b, simply by communicating infrequently with the master.

We observe similar trends in Figure 5a-5b for our asynchronous operation, where workers synchronize with the master at arbitrary time intervals as per Algorithm 2. Specifically, in our experiments, for each $r \in [R]$, the time interval for the $r$th worker is decided uniformly at random from $[H]$ after every synchronization by that worker. This ensures that $gap(\mathcal{I}_T^{(r)}) \leq H$ holds for every worker $r \in [R]$ and the schedule $\mathcal{I}_T^{(r)}$ is different for each of them.

**(a)** Training loss vs epochs

**(b)** Training loss vs $\log_2$ of communication budget

**Figure 4** Figure 4a-4b demonstrate the effect of incorporating local iterations and compare these effects across vanilla SGD, TopK-SGD, as well as SignTopK.

**(a)** Training loss with the communication budget for our schemes against baselines

**(b)** Test error using a model trained for given number of iterations, as seen in Figure 5a

**Figure 5** Figure 5a-5b demonstrate the performance of our scheme in comparison with EF-SIGNSGD [15] and TopK-SGD [4, 30] in a convex setting for asynchronous operation.

# E  Summary of Results

Combining local computations with quantization and explicit sparsification enables significantly reduced communication, resulting in a lot of bit savings. For a fixed number of local iterations $H$, we characterize the required total number of iterations $T = \Omega(\cdot)$ (see Table 1 and Table 2) after which the algorithm converges at the rates of distributed vanilla SGD. Furthermore, we also characterize the reduction in communication, in terms of the asymptotic limits of local computations, i.e., $H = \mathcal{O}(\cdot)$ (see Table 1 and Table 2).

| | | Synchronous | | |
|---|---|---|---|---|
| Objective | Rate | $H$ | $T$ |
| Smooth and non-convex | $\mathcal{O}(1/\sqrt{bRT})$ | $\mathcal{O}(\gamma T^{1/4}/(bR)^{3/4})$ | $\Omega(H^4(bR)^3/\gamma^4)$ |
| Smooth and strongly convex | $\mathcal{O}(1/bRT)$ | $\mathcal{O}(\gamma\sqrt{T/(bR)})$ | $\Omega(H^2(bR)/\gamma^2)$ |

**Table 1** Summary of results for the synchronous setting with fixed learning rate in both the smooth and non-convex case and decaying learning rate in the smooth and strongly convex case.

| Objective | Rate | Asynchronous | |
| --- | --- | --- | --- |
| | | $H$ | $T$ |
| Smooth and non-convex | $\mathcal{O}(1/\sqrt{bRT})$ | $\mathcal{O}(\sqrt{\gamma}T^{1/8}/(bR)^{3/8})$ | $\Omega(H^8(bR)^3/\gamma^4)$ |
| Smooth and strongly convex | $\mathcal{O}(1/bRT)$ | $\mathcal{O}(\sqrt{\gamma}(T/(bR))^{1/4})$ | $\Omega(H^4(bR)/\gamma^2)$ |

**Table 2** Summary of results for the asynchronous setting with fixed learning rate in both the smooth and non-convex case and decaying learning rate in the smooth and strongly convex case.