[Reviews · NeurIPS 2019]

Reviewer 1



UPDATE: I thank the authors for their detailed response to reviewer feedback. Overall I believe this is a quality paper, and would be happy to see it published at NeurIPS. This is reflected in my current assessment, which remains unchanged. Previous studies have contributed three classes of methods for avoiding gradient communication bottlenecks for distributed SGD: (1) communicating updates less frequently, (2) communicating less updates (random k or top k) and (3) communicating quantized updates. There is also a known method for keeping track of induced error in order to compensate later. The authors build upon these previous works by combining all three methods into a unified algorithm (QLSGD), showing that one can combine their benefits. They additionally show that the error compensation mechanism is more broadly applicable than (2), which is the main context it has been previously used. In order to demonstrate convergence properties of QLSGD, the authors were further required to generalize some tools and statements from prior literature, in some cases with improvements (e.g. getting rid of an unnecessary technical assumption from a NeurIPS paper last year). These generalizations are not stand-out contributions in isolation but do add to the overall strength and quality of the work. Pros: The authors are to be commended for combining both a very strong, theoretical analysis with experimental results on a practical distributed SGD problem (ResNet-50 for ImageNet). As far as I can tell their results represent a new state-of-the-art for bits transmitted in a master-based distributed SGD setup. Cons (all minor points): The authors focus entirely on worker -> master communication complexity, but never discuss the reverse direction (broadcasting updated network weights). Tackling this issue directly is not necessarily within the scope of this study, but it certainly warrants discussion. Even a 100x improvement for half the communications is "only" a 50% speed-up. Additionally, the authors should take note of this related work from ICML this year, "Decentralized Stochastic Optimization and Gossip Algorithms with Compressed Communication" (https://arxiv.org/pdf/1902.00340.pdf). This appears to already address 2/3 of what the authors propose in this study, which does detract slightly from the overall novelty. I further refer the authors to Ofer Dekel's SysML 2019 keynote, where he describes at length why claims of the form "we achieve X% cost reduction for only Y% loss in accuracy" are flawed / very difficult to interpret. I largely agree with his opinion on the matter.

Reviewer 2



*author feedback* I thank the authors for addressing my comments. I am willing to increase my score to 5 and I hope that the authors will do a thorough pass over the paper for the final version, if accepted. A better description and discussion of the experimental results must be added and I would also recommend to simplify presentation and add more intuition about the challenges faced in the analysis to the main part of the paper in order to put more focus its contributions. *summary* The authors propose and analyze a new algorithm, called Qsparse-local SGD. It combines three known concepts for achieving communication efficiency into a unified algorithm. That is sparsification, quantization and local computation. The proposed algorithm is analyzed theoretically and evaluated for training Resnet-50 on the ImageNet dataset. *comments* [Presentation] The paper is generally well written and easy to follow. I like the introduction with the related work where the differences to (closely) related work is explained in detail and effort is put into defining the contributions and limitations of the proposed method. The presentation of the paper is a bit dense and a lot of equations and enumerations are put inline. This could be improved by adding a bit more structure. An additional 9th content page could help. [Novelty] The novelty of the paper is the combination of existing techniques into one algorithm. This comes with some theoretical challenges as the authors say. Since this is the main contribution of the paper it would be nice to have more technical details on what these challenges are in particular and how they are solved. While the combination of existing methods is a valid contribution I would like to see more thorough experimental results showing the merit of the individual components that you put together w.r.t the overall performance. For example, it would be interesting to see what the gain is of using quantization on top of sparsification? How do the quantization level s and the sparsification level k play together? How would one tune these values? [Experiments] The experimental section is difficult to follow and important details are missing. This needs to be improved, in particular: - you use sign quantization, topk sparsification but what is the synchronization schedule you used to distributed the work? How many local steps are performed on each worker before synchronization? - The figures and the reference schemes need to be explained better. What do the individual curves correspond to? What do the numbers in the legend mean? Which curve corresponds to local SGD without quantification or sparsification? This has to be stated in the main part. - Did you tune the parameters for the individual methods to have a fair comparison? - Please refer to the appendix if the reader can find additional experiments or supporting material there. Minor comments: - The definition of b is hidden in the algorithm, should be somewhere in the text - In line 176 you refer to assumptions (i), (ii) whereas you use (i) for enumeration in multiple places. Use (A1) (A2) or something else for it to stand out. *Summary* The paper is generally well written apart from the experimental section which needs to be improved. The contribution consists of the combination of existing techniques and their analyses. This is fine, but for the contribution to be strong enough I think the technical challenges should be emphasized and explained better and the merit of the individual components, as well as their tradeoffs should be evaluated more carefully.

Reviewer 3



I would like to thank the authors' feedback. I hope you can add more details about your experiments in the revised version. The score will be unchanged. --------------------------------------------------------------------------------------------------------- This is a good work and the authors are quite familiar with this research area, but the experiment evaluation is not adequate. I have several questions: 1. In algorithm 1, why x_0 and \hat{x}_0^{(r)} are all initialized as 0 ? x_0 and \hat{x}_0^{(r)} are all model parameters, setting them to 0 is unreasonable. 2. In the experiment part, what doest SignTopK_4L mean? You should describe your experiments in more detail. 3. More experimental results should be given, such as sync & async comparison , the effect of different sparsification strategies or quantization mehtods.

Reviewer 4



a). My first concern is on the novelty. While the paper does a good combination of existing techniques, none of any individual part is interested enough. - The motivation to combine these techniques are straightforward, authors do not show many insights and interesting things on this point. - Thus, I expect more interesting things can be discovered from the mathematic analysis. For example, whether authors can propose some usage of new analysis methods/tools, or strong results comparing with existing ones. - While authors have done a really solid analysis for the proposed work, I fail to find the desired interesting points, which I have mentioned above. b). The presentation can be improved. - I do not see many new things in Section 3.2 and 4.1, these sections are not necessary. - Authors give a summary in Appendix.E, I hope they can move this table to the main text. Besides, it is better to add a comparison between related methods used in the experiments. This can help readers understand how tight the given bound is compared with other distributed SGD methods. - Appendix D should be moved in the main text in Section 5, they are necessary for ablation studies. Deep models are indeed popular, but it is not a good testbed to verify the obtained theoretical results as assumptions can be hardly satisfied.

[Author Response · NeurIPS 2019]

**General comments:** • We will make our implementation open source, if this work gets accepted. • $\mathbf{x}_0$ and $\widehat{\mathbf{x}}_0^{(r)}$ can be randomly set to the same point and our analysis does not require them to be $\mathbf{0}$; we can change this, if accepted. • We will define mini-batch size $b$ in the main text of the final version, if accepted. • While the aggregate broadcast has not been our focus, as in [4, 19, 38], it can be inexpensive when the broadcast routine is implemented in a tree-structured manner as in many MPI implementations, or if the parameter server aggregates the sparse quantized updates and broadcasts it. We will include a discussion if accepted. • We have compared the communication budget required for achieving a given target accuracy across different techniques and demonstrate that *Qsparse-local-SGD* is the most efficient as it improves on the Pareto frontier. Therefore, we are not making an absolute statement, but a comparison between our scheme and the state-of-the-art, thereby, not contradicting the views of the speaker.

**Description of technical challenges:** As discussed in lines 87 to 90 of the paper, the analysis of *Qsparse-local-SGD* poses several technical challenges, including (i) the proof of controlled evolution of the error as well as the deviation of the local iterates, and (ii) asynchronous updates together with distributed compression using operators which satisfy Def 3, including our composed (*Qsparse*) operators. To the best of our knowledge, all the results in the paper are new and there are several technical challenges to prove each of them; we highlight a small selection of them below.

From literature [3, 29], we know that methods with error compensation work only when the evolution of the error is controlled. Unlike previous works, *Qsparse-local-SGD* stores the compression error of the net *local update*, which is a sum of at most H gradient steps and the historical error, in the local memory. Our analysis of the controlled evolution of memory while using any compression operator satisfying Def 3 required some work; see Appendix B.3. Furthermore, we also need to prove that the local iterates which evolve on their own do not arbitrarily deviate from each other; analysis of which is provided in Appendix B.4. Another useful technical observation is that the composition of a quantizer and a sparsifier results in a contraction operator (Def 3); see Appendix A for proofs on the same.

This gets more involved in the asynchronous setting as discussed in line 274. In such a scenario, the proof of the average true sequence being close to the virtual sequence is non trivial (Appendix C.5, C.6) and requires carefully chosen reference points on the global parameter sequence lying within bounded steps of the local parameters. This problem also arises in bounding the deviation of local iterates, the details of which is provided in Appendix C.3, C.4.

**Experiments:** We do have comparisons between the individual techniques of quantization, sparsification, local iterations, and their smaller combinations, with *Qsparse-local-SGD*. Our preliminary experiments demonstrated superior performance of $Sign$ operator composed with TopK-SGD, as compared to other quantizers and sparsifiers. Therefore, we use the $Sign$ operator as our quantizer and TopK as the sparsifier. Our main objective through numerics has been to demonstrate that our algorithm outperforms the cases when quantization, sparsification, and local iterations are being used individually or in pairs. In Fig. 1c, we observe that SGD with 4 local iterations requires $2.83\times$ less bits than vanilla SGD to achieve $70\%$ top-1 accuracy, EF-signSGD and TopK-SGD require $16\times$ and $128\times$ less bits, respectively, for the same. Therefore, both quantization and sparsification, when individually used with error feedback, are superior to vanilla SGD with or without local iterations. When these techniques are all put together, we have SignTopK-SGD with 8 or 16 local iterations, which demonstrates more than $1024\times$ gain over vanilla SGD and is superior to all the above methods. We observe similar trends in the Appendix in Fig. 4, for top-5 accuracies, and in Fig. 2, which runs the synchronous operation, for a convex objective. We also have comparisons for the asynchronous operation in Fig. 3 in Appendix D.

We wish to clarify that Fig. 1-4 compare the performance of our composed (*Qsparse*) operator, namely SignTopK, with (i) $Top_k$ SGD with error compensation (TopK), (ii) SignSGD with error compensation (EF-signSGD), and (iii) vanilla SGD (SGD). All of these are specializations of *Qsparse-local-SGD*. Furthermore, SignTopK uses a synchronization period of 1, whereas SignTopK_4L uses a synchronization period of 4, similarly for _8L, _16L.

**Parameter tuning:** *Qsparse-local-SGD* has a lot of flexibility and many different specializations, which can be realized by using local iterations with different combinations of quantizers and sparsifiers (in a piecewise manner). One way of tuning the number of quantization levels ($s$) and sparsification parameter ($k$) numerically is that, for a given $\gamma$ and $H$, find an $(s, k)$ from the family of operators corresponding to the $\gamma$ that minimizes the total number of bits. $H$ can be varied without disturbing the proposed asymptotic limits of local iterations, see Corollary 1, 2 and Theorem 3, 4.

**Comparison with paper on decentralized optimization [15]:** We had given a brief distinction of our work with [15] in footnote 2 of the paper, which we expand below. The main distinctions of *Qsparse-local-SGD* from [15] are (i) our algorithm employs local iterations with both synchronous and asynchronous operations, (ii) provides guarantees for both smooth (non-convex) and strongly convex objectives, (iii) combines quantization and sparsification. [15] uses operators satisfying Def 3, which could be either a quantizer or sparsifier (not both) with focus on decentralized SGD *without* local iterations and *only* for strongly convex objectives. In contrast, we propose a class of composed operators (*Qsparse*) satisfying Def 3, (which is directly useful in [15]), which is applied infrequently onto the net local updates before synchronization. Our analysis also results in characterization of asymptotic limits of local computations for general compression operators. Therefore, to the best of our understanding, the results in the paper are not covered in [15].

[Meta-Review · NeurIPS 2019]

The authors propose an algorithm for distributed SGD that combines gradient sparsification, gradient quantization and local computation with periodic gradient accumulation. Novelty may be limited (as a combination of existing techniques), but the authors claimed that the theoretical analysis is challenging, and experimental results are reported on a practical distributed SGD problem. The paper is generally well written and easy to follow. However, the main concern is on the the experimental section, which is difficult to follow and important details are missing. A better description and discussion of the experimental results should be added.